# Non-selective inhibition of inappropriate motor-tendencies during response-conflict by a fronto-subthalamic mechanism

Jan R Wessel[1,2]*, Darcy A Waller[2], Jeremy DW Greenlee[3]

[1]Department of Neurology, University of Iowa Hospitals and Clinics, Iowa City, United States; [2]Department of Psychological and Brain Sciences, University of Iowa, Iowa City, United States; [3]Department of Neurosurgery, University of Iowa Hospitals and Clinics, Iowa City, United States

**Abstract** To effectively interact with their environment, humans must often select actions from multiple incompatible options. Existing theories propose that during motoric response-conflict, inappropriate motor activity is actively (and perhaps non-selectively) suppressed by an inhibitory fronto-basal ganglia mechanism. We here tested this theory across three experiments. First, using scalp-EEG, we found that both outright action-stopping and response-conflict during action-selection invoke low-frequency activity of a common fronto-central source, whose activity relates to trial-by-trial behavioral indices of inhibition in both tasks. Second, using simultaneous intracranial recordings from the basal ganglia and motor cortex, we found that response-conflict increases the influence of the subthalamic nucleus on M1-representations of incorrect response-tendencies. Finally, using transcranial magnetic stimulation, we found that during the same time period when conflict-related STN-to-M1 communication is increased, cortico-spinal excitability is broadly suppressed. Together, these findings demonstrate that fronto-basal ganglia networks buttress action-selection under response-conflict by rapidly and non-selectively net-inhibiting inappropriate motor tendencies.

DOI: https://doi.org/10.7554/eLife.42959.001

*For correspondence:
jan-wessel@uiowa.edu

## Introduction

Rapid action-selection is paramount to effective goal-directed behavior. Many real-life situations simultaneously activate multiple, often incompatible motor tendencies, thus resulting in response-conflict (*Botvinick et al., 2001*; *Simon and Rudell, 1967*; *Stroop, 1935*; *Yeung et al., 2004*). To maintain successful behavior, the human brain needs a mechanism to rapidly resolve such response-conflict. Most early research implicated the medial prefrontal cortex (mPFC) in its detection and processing (*Garavan et al., 2003*; *Taylor et al., 2007*; *Ullsperger and von Cramon, 2001*; *van Veen et al., 2001*). Subsequent research has extended this picture to include the basal ganglia (BG, *Brittain et al., 2012*; *Frank et al., 2007*; *Zavala et al., 2013*; *Zavala et al., 2014*). Current models of action-selection hold that controlled, non-automatic motor activity is regulated via dynamic fronto-BG interactions (*Hikosaka and Isoda, 2010*; *Wessel and Aron, 2017*; *Wiecki and Frank, 2013*). However, it is unclear how exactly these fronto-BG networks mechanistically exert influence on the motor system during response-conflict.

Here, we tested the theory that motoric response-conflict invokes a specific fronto-BG neural mechanism for inhibitory control (*Frank, 2006*; *Wessel and Aron, 2017*). We propose that this mechanism broadly and rapidly net-inhibits inappropriate motor tendencies during motoric response-conflict, thereby facilitating the selection of the correct response. The proposed mechanism is implemented via a fronto-BG network, which includes the mPFC and the subthalamic nucleus

(STN) in the BG. Specifically, it purportedly involves a hyper-direct inhibitory pathway from cortex to STN (*Aron and Poldrack, 2006*; *Mink, 1996*; *Nambu et al., 2002*), which can rapidly inhibit thalamocortical motor representations via the output nuclei of the BG (globus pallidus pars internus, substantia nigra pars reticulata) and the thalamus (*Schmidt et al., 2013*; *Wessel and Aron, 2017*). The inhibitory control function of this mechanism is well-characterized and is typically studied in the stop-signal task (SST, *Aron and Poldrack, 2006*; *Logan and Cowan, 1984*), where it enables the rapid outright cancelation of impending motor responses after an explicit signal to stop a response (*Aron et al., 2014*; *Jahanshahi et al., 2015*; *Richard Ridderinkhof et al., 2011*). We propose that this same mechanism is part of a cognitive control cascade that is recruited during motoric response-conflict and serves to inhibit inappropriate responses in favor of the correct response.

During outright action-stopping, the activity of this mechanism can be measured in several ways. Cortically, fronto-central low-frequency activity (~2–8 Hz) is increased during action-stopping (*Huster et al., 2013*; *Nigbur et al., 2011*; *Yamanaka and Yamamoto, 2010*), likely reflecting activity of the mPFC (*Huster et al., 2011*). According to popular models, the mPFC then recruits the subcortical STN to net-inhibit the motor cortex (*Jahanshahi et al., 2015*; *Rae et al., 2015*; *Wiecki and Frank, 2013*). In line with this, successful action-stopping is accompanied by increased STN activity (*Aron and Poldrack, 2006*; *Benis et al., 2014*; *Benis et al., 2016*; *Kühn et al., 2004*; *Ray et al., 2012*). Interestingly, unlike the 2–8 Hz increase observed at fronto-central scalp sites, stopping-related activity in STN occurs largely in the β frequency-band (13–30 Hz, *Benis et al., 2014*; *Ray et al., 2012*; *Wessel et al., 2016*) (*Aron et al., 2016*). Finally, at the motor system level, the net-inhibitory influence of this mechanism is expressed as a non-selective reduction in cortico-spinal excitability (CSE). In other words, action-stopping via this mechanism leads to CSE suppression of both task-related and task-unrelated muscles (*Badry et al., 2009*; *Cai et al., 2012*; *Majid et al., 2012*; *Wessel et al., 2013b*). Such non-selective CSE-suppression during stopping is directly related to STN β-band activity (*Wessel et al., 2016*), which provides a link between the neural activity of the fronto-BG inhibitory control mechanism and the physiological manifestation of inhibition of the motor system.

The hypothesis that this same motor inhibition mechanism is also recruited during response-conflict is based on several observations. First, response-conflict leads to a slowing of motoric response times (*Simon and Rudell, 1967*; *Stroop, 1935*), which some propose to result from active inhibition (*Burle et al., 2002*; *Ridderinkhof, 2002*). Second, just like action-stopping, response-conflict is commonly accompanied by low-frequency fronto-central scalp-recorded activity (*Cavanagh and Frank, 2014*; *Cohen and Cavanagh, 2011*; *Nigbur et al., 2012*). Third, computational accounts of cognitive control in the BG propose a key role for the STN in regulating behavior during response-conflict (*Frank, 2006*; *Wiecki and Frank, 2013*). Fourth, STN recordings have shown increases in both low-frequency (2–8 Hz) and β-band activity during response-conflict (*Brittain et al., 2012*). However, to date, no evidence for the broad, non-selective suppression of the motor system during response-conflict exists. Moreover, direct evidence for interactions between the BG and motor cortex during conflict is hard to obtain in humans, as simultaneous recordings of subcortical BG activity and cortical motor activity are extremely rare.

Here, we present a multi-modal investigation that included three experiments designed to test the hypothesis that the STN of the BG non-selectively suppresses motor cortex during response-conflict. Experiment 1 (single-trial scalp-EEG) tested whether potential behavioral expressions of motor inhibition during response-conflict (viz., the slowing of response times) are related to ubiquitous EEG signatures that are commonly found during outright action-stopping – which could indicate the presence of a generic cognitive control mechanism during conflict. Since Experiment 1 indicated that action-stopping and motor slowing due to response-conflict are related to a shared signature on the scalp, we then tested whether the purported inhibitory influence that the subcortical STN exerts onto active motor representations in M1 during action-stopping is also observable during response-conflict. To this end, in Experiment 2, we then performed unique simultaneous intracranial recordings of STN and M1 activity in awake, behaving neurosurgical patients. Using measures of functional and effective connectivity, we tested whether response-conflict leads to increased communication between STN and the specific parts of M1 that represent inappropriate, to-be-inhibited motor tendencies (in line with the purported inhibitory role of the STN). Since this was found to be the case in Experiment 2, in a final step, we then tested whether response-conflict leads to a non-selective suppression of cortico-spinal excitability – a unique signature of motor suppression

observed during outright action-stopping. To that end, Experiments 3.1. and 3.2. used transcranial magnetic stimulation to test whether the net-excitability of the motor system is broadly and non-selectively suppressed during response-conflict. Taken together, these studies aimed to provide converging evidence for the fact that action-stopping and response-conflict involve overlapping neural mechanisms, observable both on the scalp and subcortical levels, with comparable outright effects on the net-excitability of the motor system. This would demonstrate that response-conflict is partially resolved by a neural mechanism for non-selective motor inhibition.

## Results

### Inhibitory behaviors during response-conflict share EEG signature with outright stopping

In Experiment 1, 21 healthy adult participants performed a visual Simon task (*Simon and Rudell, 1967*) to elicit response-conflict (*Figure 1A*) and a stop-signal task (SST, *Logan and Cowan, 1984*) to elicit outright action-stopping (*Figure 1B*). During both tasks, participants showed the expected behavior. In the Simon task, correct incongruent trial reaction times were slower compared to congruent trial reaction times (361 ms [SEM: 8] vs. 326 ms [SEM: 7]; $t(20) = 9.63$, $p=5.92*10^{-09}$, $d = 1.06$), and error rates on incongruent trials were significantly increased compared to congruent trials (22.67% vs. 10.38%; $t(20) = 4.6$, $p=0.0002$, $d = 1.4$), indicating response-conflict. In the stop-signal task, correct-trial RT was slower compared to failed-stop trial RT (551 ms vs. 469 ms; $t(20) = 14.45$, $p=4.77*10^{-12}$, $d = 1.44$), indicating adherence to the race-model of the SST (*Logan and Cowan, 1984*; *Verbruggen and Logan, 2008*). Mean stop-signal RT was 238 ms, and mean stopping success rate was 52%, indicating convergence of the adaptive stop-signal delay algorithm (which aimed to keep successful stopping rates around 50%).

Throughout both tasks, we non-invasively recorded scalp-EEG. The SST was used to elicit a well-known neural scalp-signature of motor inhibition. This signature was then isolated from the EEG signal mixture using independent component analysis (ICA). We then tested whether this component exhibits increased activity during response-conflict in the Simon task, and whether this activity relates to purported behavioral indices of inhibition (motor slowing) on a trial-to-trial basis.

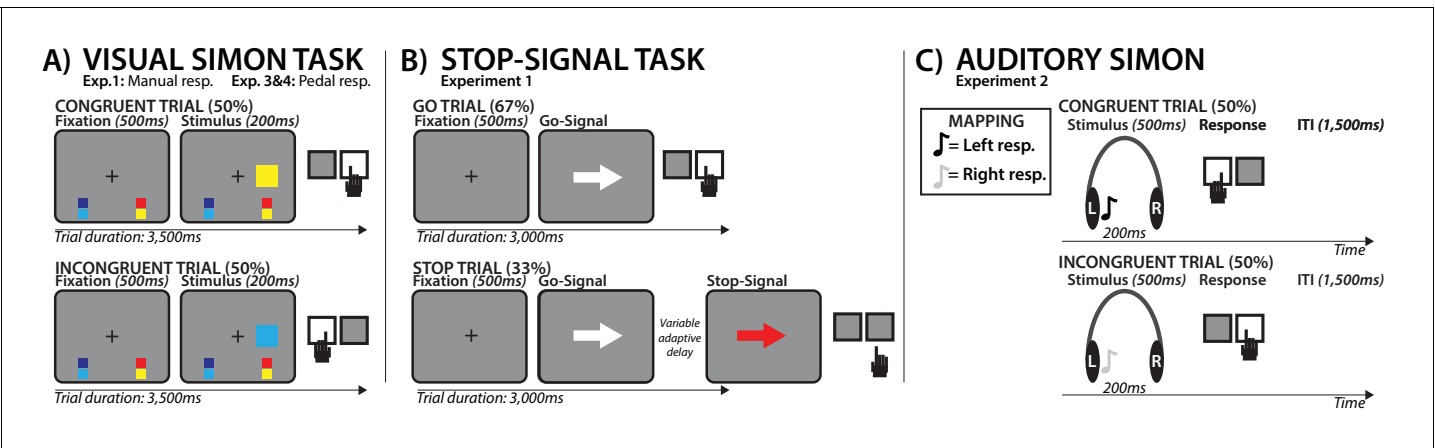

**Figure 1.** Task diagrams for all experiments. (**A**) Visual Simon task. Fixation was followed by a colored square, with color indicating the required response hand (according to the response mappings, which were displayed on the bottom of the screen throughout the experiment). Response hand either matched the side of stimulus presentation (congruent trial) or did not match (incongruent trial). Incongruent trials introduced response-conflict. In Experiment 1 (EEG), responses were made with the hands, whereas in Experiment 3.1. and 3.2. (TMS and EMG) they were made with the feet so that the hand muscles were task-unrelated. (**B**) Visual stop-signal task used in Experiment 1 to evoke neural signature of motor inhibition. (**C**) Auditory Simon task, used in Experiment 2. Intraoperative recordings were performed on participants with partially occluded vision, therefore the experiment was conducted entirely in the auditory domain. A high-frequency tone indicated a right hand response, a low frequency tone indicated a left hand response. Response-conflict was created through incongruent laterality of the stimulus presentation, same as in the visual experiment used in Experiments 1, 3.1., and 3.2.

DOI: https://doi.org/10.7554/eLife.42959.002

ICA (*Jutten and Herault, 1991*) was used to disentangle the individual neural source signals underlying the scalp EEG mixture (*Onton et al., 2006*). ICA is designed to separate neural sources that independently contribute to the compound scalp-EEG signal mixture. For example, ICA can disentangle motor, sensory, and cognitive processes, even when they are simultaneously active (*Makeig et al., 1996*), and it can even disentangle sources that have overlapping scalp-distributions and frequency-properties when present in the data (e.g., two independent fronto-medial theta-band components can be disentangled with as few as 31 electrodes/data dimensions [*Makeig et al., 2004*]). Hence, a combined ICA was performed on every subject's SST and Simon task EEG data.

We used only the SST portion of the data as a 'functional localizer' to identify one independent component (IC) per subject from the combined ICA. This component was selected to reflect the known properties of neural activity related to inhibitory control in the stop-signal task. We used the following criteria to select this component (for more details, see the Materials and methods section of this paper and *Wessel and Aron, 2015*): First, the IC showed a fronto-central scalp distribution of the component weight matrix. Second, the IC showed an earlier onset of its P3 event-related potential (P3-ERP) following stop-signals when actions were successfully stopped compared to when commission errors occured (this property reflects the race-model of the SST, which holds that an earlier onset of the stopping-process will lead to more successful stopping). Third, across subjects, there was a positive correlation between the onset of the selected ICs' stop-signal P3-ERPs and each subject's stop-signal reaction time (which illustrates the relationship between the timing of the neural process captured in this component and the speed of the motor inhibition process).

ICs were selected automatically using algorithms that operationalized these criteria (see Materials and methods). *Figure 2A,B* show that component selection was successful, as the thusly selected components fulfilled the above-mentioned criteria. *Figure 2A* also shows the time-frequency decomposition of the components, demonstrating that their event-related activity after stop-signals is dominated by the lower frequency-bands (2–8 Hz). Taken together, these components show all properties commonly observed for the purported neural scalp index of motor inhibition in the SST (*Kenemans, 2015*; *Kok et al., 2004*; *Wessel and Aron, 2015*).

We then investigated the activity of these components during the Simon task in each subject's EEG data to test our two hypotheses. In line with our first hypothesis, we found significantly (p<0.01, FDR-corrected) increased activity in the selected components on incongruent compared to congruent Simon trials (*Figure 2C*, left). Moreover, this activity took place in the same frequency bands that dominated those components' activity in the SST (2–8 Hz, *Figure 2A*). To test the second hypothesis (association between neural signal and purported trial-to-trial behavioral indices of inhibition in the Simon task), a single-trial, sample-to-sample time-frequency regression analysis was then performed on incongruent trials only, for each subject separately (with the resulting beta-weights tested for significance on the group-level). This revealed that the degree of low-frequency activity on individual incongruent trials in the Simon task was directly related to the degree of motor slowing on the same trial (*Figure 2C*, right). Specifically, individual incongruent trials with greater low-frequency activity in the selected 'motor inhibition' component also showed increased motor slowing (*Figure 2D*).

Taken together, the results of Experiment one suggest that a similar fronto-central low-frequency scalp signature is present during both outright action-stopping (in the stop-signal task) and during conflict-related motor slowing (in the Simon task). Moreover, ICA could not disentangle both processes, suggesting that they may share a common neural source signal generator (though brain lesion work would be ultimately needed to confirm this hypothesis). Importantly, this low-frequency activity directly related to trial-by-trial behavioral indices of motor inhibition in both task contexts: its timing delineated successful from failed stopping in accordance with the race-model of the stop-signal task, and its activity on incongruent Simon trials directly relates to the degree of motoric slowing on each trial.

At a minimum, Experiment 1 indicates that behaviors that reflect motor inhibition (stopping and slowing) relate to common fronto-central scalp signatures across both tasks. While these observations are in line with the interpretation that response-conflict may be resolved via fronto-medially initiated inhibitory control, Experiment 1 on its own cannot prove this assertion.

Indeed, if response-conflict elicits net-motor inhibition mediated by a fronto-BG neural mechanism (in which mPFC triggers BG activity that ultimately net-inhibits M1), increased communication between the BG (specifically, the STN) and the motor system during response-conflict would have to

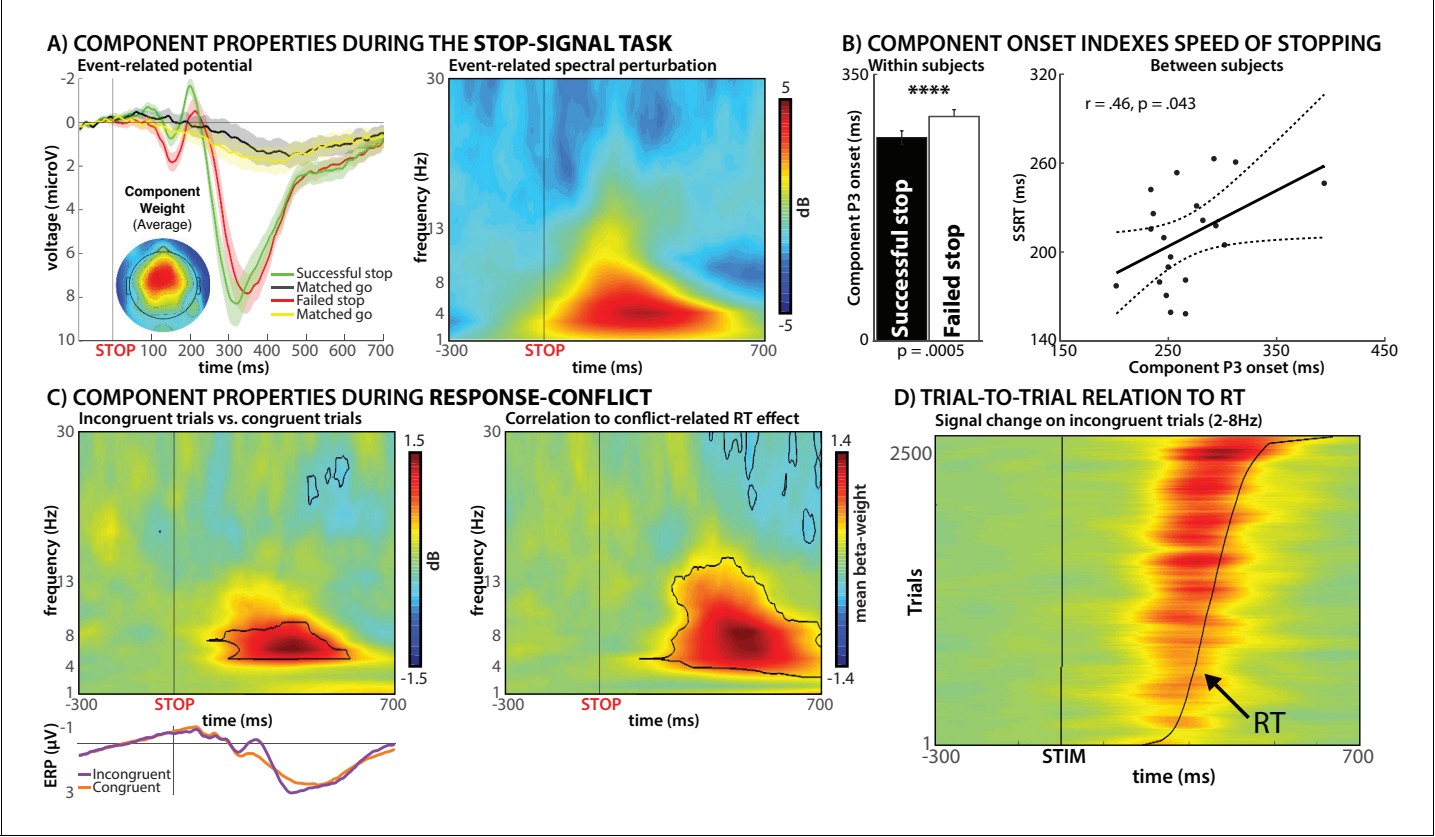

**Figure 2.** Results from Experiment 1. (**A**) Group-average event-related potential and event-related spectral perturbation of the selected independent EEG source-signal components during the stop-signal task. The left panel shows the event-related stop-signal P3 activity of these components, while the right panel shows that the bulk of the signal increase of these components during successful stopping is found between 2 and 8 Hz. (**B**) Relationship between the onset of the event-related part of the components in panel A to behavioral measures of motor inhibition in the stop-signal task. The onset of the stop-signal P3 occurred significantly earlier in successful vs. failed stop-trials (left panel). Moreover, the latency of P3 onset was positively correlated with the speed of stopping across subjects (right panel). (**C**) Activity of these same components during response-conflict in the Simon task. The left panel shows a significant (p<0.01, FDR-corrected) increase in low-frequency activity on incongruent vs. congruent trials on the group-level. The right panel shows the results of a trial-by-trial correlation between the degree of motor slowing on each individual incongruent trial and the activity of the selected components, revealing a positive relationship between the same low-frequency activity and motor slowing during response-conflict. (**D**) Illustration of that same relationship on individual trials, stacked across all subjects (for visualization purposes only). Y-Axis shows individual incongruent trials, sorted by reaction time (curved black line). Stronger low-frequency activity can be observed on trials with more motor slowing (top of graph).
DOI: https://doi.org/10.7554/eLife.42959.003

be found. Moreover, a broad net-suppression of cortico-motor excitability should take place during response-conflict, just as is found during action-stopping. Therefore, following Experiment 1, Experiment 2 tested whether STN communicates with the motor system during response-conflict in a fashion that is consistent with an inhibitory influence of STN onto M1, while Experiments 3.1. and 3.2. tested whether true physiological suppression of the motor system can be observed during response-conflict.

## Experiment 2: STN influences M1-representations of inappropriate response tendencies

In Experiment 2, we intraoperatively recorded local field potentials from the STN in nine Parkinson's patients undergoing deep-brain electrode implantation surgery during performance of an auditory Simon task (*Figure 1C*). Simultaneously, we recorded activity from the bilateral primary motor cortices (M1) via subgaleally implanted strip electrodes. Detailed symptom information for each patient can be found in *Table 1*.

**Table 1.** Symptom information for each patient.

UPDRS: Unified Parkinson's Disease Rating Scale (Part III). Hemibody: L = symptoms predominantly on the left, R = right. Subtype: A/R = akinetic/rigid, T = tremor. Hand: R = right handed, L = left handed. Burrhole placement: cm lateral from midline. Selected bipolar contact: 3 = most anterior pair, 2 = middle pair, 1 = most posterior pair.

| Patient | UPDRS score | | PD subtype | | Demographics | | Burrhole placement (cm) | | Selected bipolar contact (1-3) | |
|---|---|---|---|---|---|---|---|---|---|---|
| # | OFF | ON | Hemibody | Subtype | Age (yrs) | Handed | Left | Right | Left | Right |
| 1 | 74 | 57 | L = R | A/R | 48 | R | - | - | 2 | 2 |
| 2 | 41 | 21 | L > R | A/R | 60 | R | 3.4 | 4.2 | 3 | 1 |
| 3 | 59.5 | 9 | R > L | T | 70 | R | 4.3 | 4.9 | 3 | 2 |
| 4 | 45 | 18 | L > R | T | 57 | L | - | - | 2 | 2 |
| 5 | 36 | 10 | L > R | T | 66 | R | 4.5 | 4.5 | 1 | 2 |
| 6 | 39 | 4 | R > L | T | 65 | R | - | - | 3 | 2 |
| 7 | 38 | 16 | R > L | T | 70 | R | 4.5 | 3.3 | 3 | 1 |
| 8 | 53.5 | 22.5 | R > L | T | 63 | R | 4.4 | 3.9 | 2 | 2 |
| 9 | 29 | 0 | L = R | A/R | 42 | R | 3.5 | 4 | 2 | |

DOI: https://doi.org/10.7554/eLife.42959.004

Patients showed the expected behavior in the Simon task. Correct incongruent trial reaction times were slower compared to congruent trial reaction times (763 ms [SEM: 22] vs. 708 ms [SEM: 36]; t(8) = 3.23, p=0.012, d = 0.66), and error rates on incongruent trials were significantly increased compared to congruent trials (22.84% vs. 13.54%; t(8)=3.93, p=0.0044, d = 0.92), indicating the presence of response-conflict.

Before analyzing the LFP data, we confirmed accurate electrode contact locations for both STN and M1. For STN, bipolar montages showed the expected β-band firing pattern during rest, which is used intraoperatively to confirm electrode placement for post-operative clinical stimulation (*Figure 3*).

For subgaleal M1, we identified bipolar montages that showed the expected movement-related β-band desynchronization (*Pfurtscheller and Lopes da Silva, 1999*) time-locked to the response (*Figure 4*). This validation was based on an independent peri-operative experimental session that did not involve incongruent stimuli (see Materials and methods).

To test our hypothesis regarding the interaction between STN and M1 during conflict, we then quantified inter-site connectivity using phase-locking value (PLV, *Lachaux et al., 1999*), a frequency-resolved measurement of instantaneous phase-synchrony between two remote brain regions, which indicates functional connectivity (*Fries, 2005*). PLV quantifies inter-regional interactions using the frequency-specific phase angle at each site and is independent of signal power. We were specifically interested in the interactions between STN and the M1 representation of the incorrect response tendency (i.e., the M1 hand-representation *ipsilateral* to the correct response hand). On incongruent trials, this is the motor representation that needs to be inhibited to enable correct responding. In line with this, PLV between STN and ipsilateral M1 was increased on incongruent compared to congruent trials, specifically in the β frequency-band (*Figure 5A*, left and middle). PLV was significantly increased between 180–240 ms following stimulus onset, with significant frequencies ranging from 18 to 22 Hz. The peak significance of this time-frequency window was p=0.00076 (d = 2.08), and increases in STN-M1 connectivity during this window were found in each individual subject (*Figure 5A*, right). We also analyzed PLV between STN and contralateral M1 (i.e., the M1 hand-representation of the correct response that does not need to be inhibited). This revealed no significant condition differences (*Figure 5—figure supplement 1*).

Notably, PLV does not allow inferences regarding the directionality of functional connectivity. Therefore, we performed a Granger-prediction analysis on the STN-M1 data on incongruent trials, which quantifies effective connectivity and allows such inferences. The Granger analysis also shows increased STN-ipsilateral M1 β-band connectivity during response-conflict. Moreover, the directional analysis shows that this is attributable to a Granger-'causal' influence of STN onto M1, whereas no

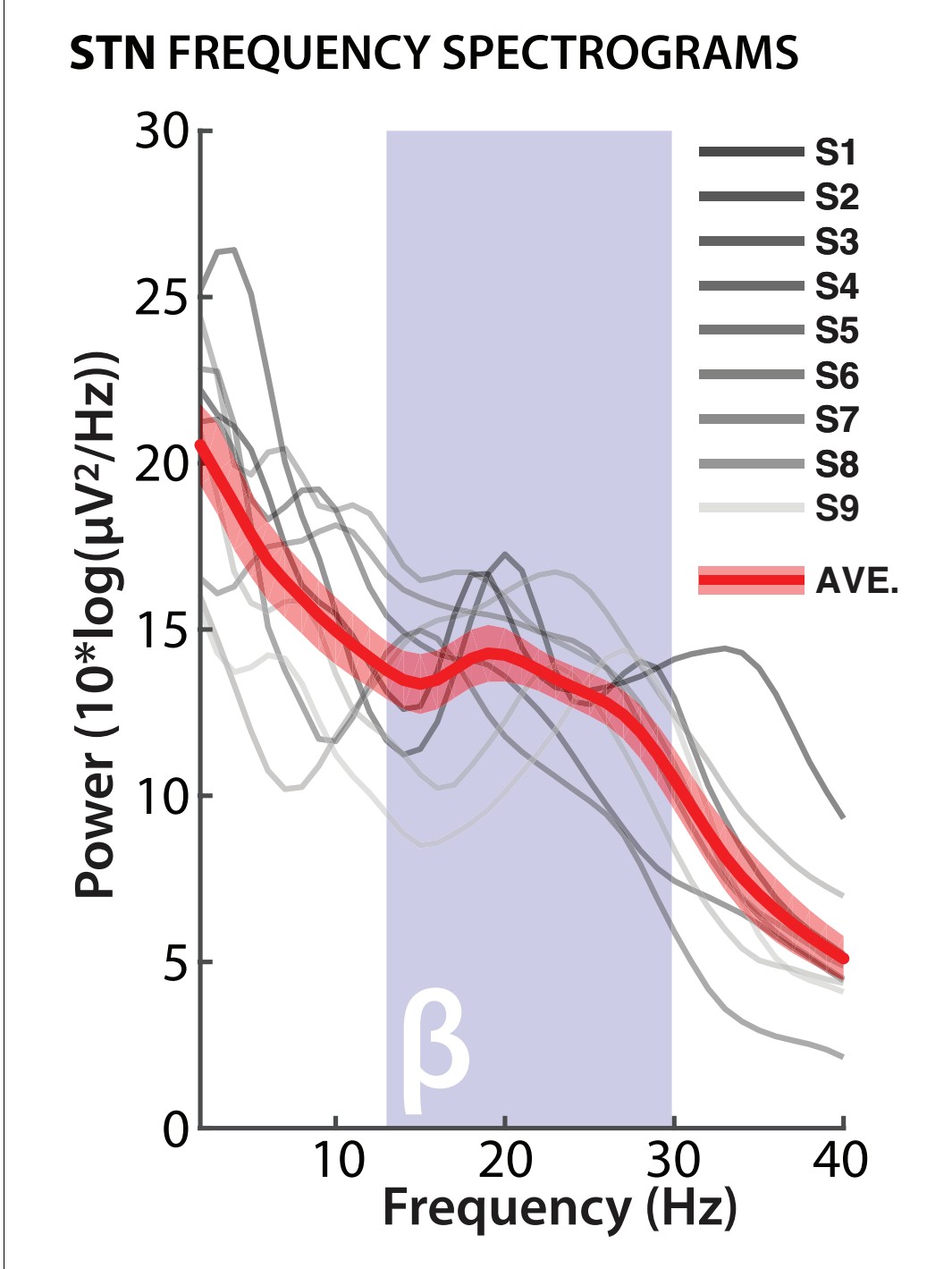

**Figure 3.** Confirmation of STN electrode placement. Bipolar STN electrode montages from selected electrodes show power peaks in β-band for each individual subject; red line shows average across all subjects (red shade: standard error of the mean).
DOI: https://doi.org/10.7554/eLife.42959.005

reverse influence is observable (*Figure 5B*). While the purpose of the Granger analysis was to investigate the directional nature of the functional connectivity identified by PLV in the significant time-frequency window, the calculation of time/frequency-resolved connectivity was done independently of the PLV analysis (and using different methods, see Materials and methods). Nevertheless, the significant STN->M1 Granger-prediction effect notably matched the PLV effect in both time and

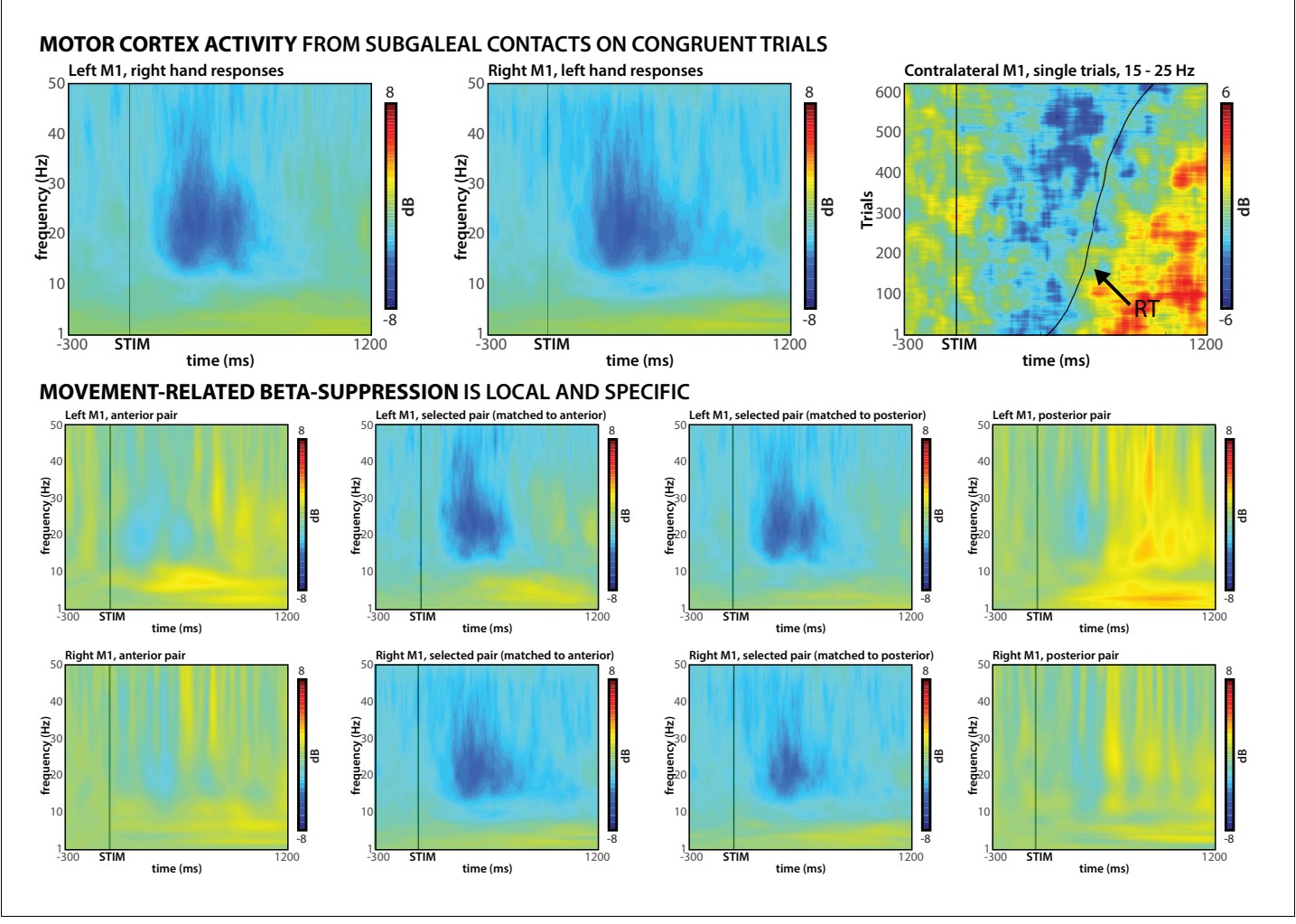

**Figure 4.** Confirmation of subgaleal M1 electrode placement. Top row: Bipolar subgaleal electrode contact montages over M1 show clear motor signatures; β-desynchronization over contralateral M1 on congruent trials can be observed in both left M1 (left) and right M1 (middle). Furthermore, single-trial data show clear alignment of this desynchronization to the response, followed by post-response β-rebound (right). Middle and bottom row: β-desynchronization is localized to the selected montages, as signatures are largely reduced/absent at anterior/posterior montages.
DOI: https://doi.org/10.7554/eLife.42959.006

frequency, providing quantification-independent evidence for neural connectivity in the β-band around 200 ms following stimulus onset. The peak significance of STN->M1 Granger-influence in this time-frequency window was p=0.0022, and increases in effective STN->M1 connectivity during this window were found in all but one individual subject (said subject also had by far the lowest PLV condition difference in the PLV functional connectivity analysis).

Taken together, the results of Experiment 2 show that functional connectivity between STN and ipsilateral M1 is increased during response-conflict. Granger analyses further suggest that this connectivity is directional, from STN to M1. This suggests that during response-conflict, the conflicting, incorrect response tendency in ipsilateral motor cortex is subject to influences of the STN (most likely via the STN's downstream connection to the motor system via the output nuclei of the BG and the thalamus; see Discussion).

## Experiment 3.1.: Response-conflict leads to non-selective CSE-suppression

While Experiment 2 provides unique and novel evidence for the fact that STN-activity influences the specific part of M1 that represents the incorrect response-tendency during motoric conflict, the

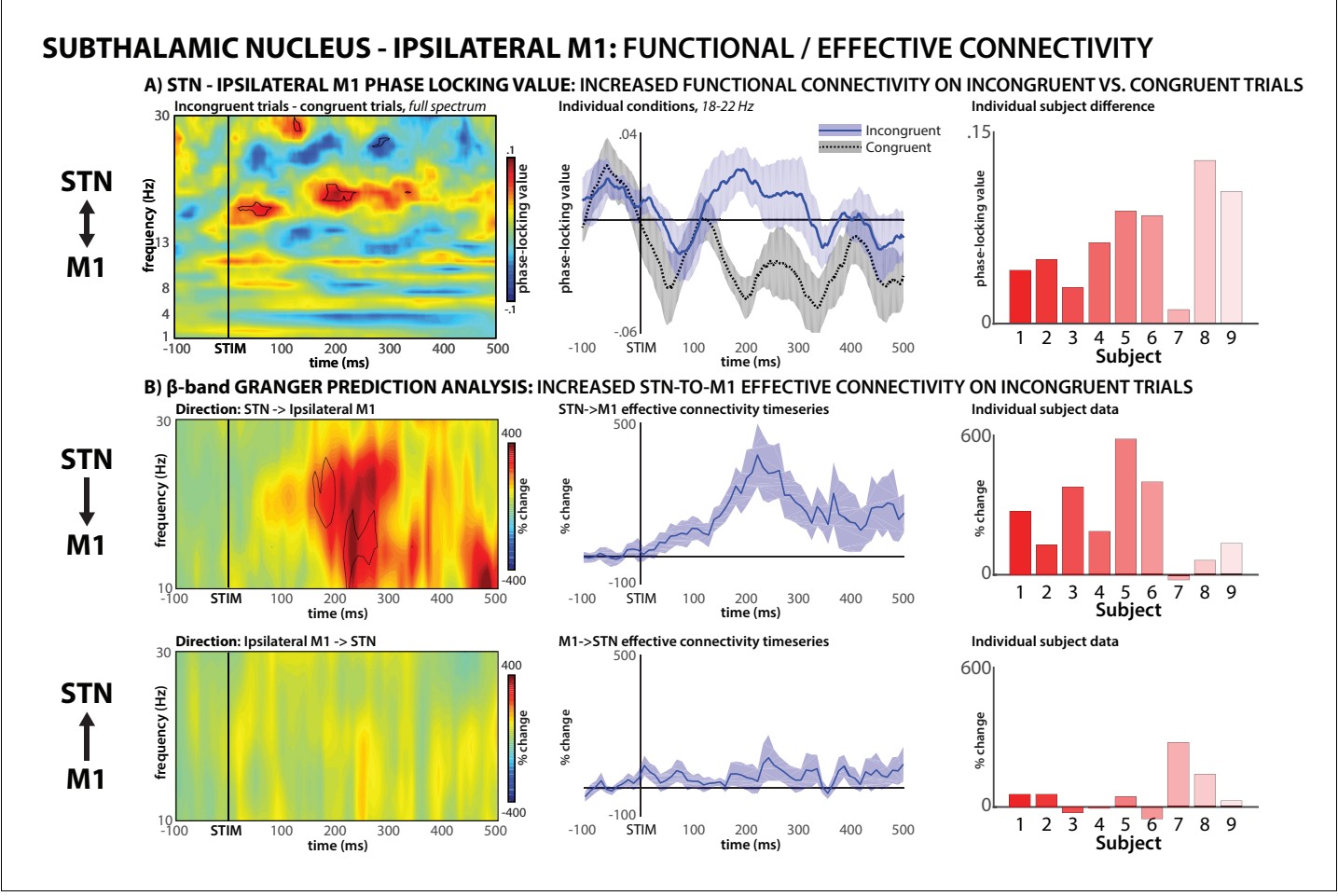

**Figure 5.** STN – M1 functional and effective connectivity during conflict. (**A**) Functional connectivity, phase-locking value. Left panel shows the full-spectrum PLV analysis, specifically, a comparison of incongruent minus congruent trials. Significant (p<0.01) group-differences between conditions are highlighted in black outline. To visualize the condition differences in the significant frequency-range, the middle panel shows individual condition data in the significant beta frequency-range, which was identified from the full spectrum analysis (18–22 Hz). Right panel shows individual subject condition differences (positive numbers indicate greater PLV in the incongruent condition). (**B**) Effective connectivity (Granger prediction). Top row shows directional influence of STN activity onto ipsilateral M1, bottom row shows the reverse direction. Left panels show full beta-spectrum analysis, significant (p<0.01) increases in directional connectivity (from baseline) are highlighted in black. Middle panels show the effective connectivity time-series at the significant frequencies identified for STN->M1 connectivity from the full-spectrum analysis. Right panels show individual subject data for the significant time-frequency window.

DOI: https://doi.org/10.7554/eLife.42959.007

The following figure supplement is available for figure 5:

**Figure supplement 1.** STN – contralateral M1 connectivity during conflict.

DOI: https://doi.org/10.7554/eLife.42959.008

presence of physiological inhibition of the motor system itself has yet to be demonstrated. Based on prior findings from the SST (*Badry et al., 2009*; *Cai et al., 2012*; *Majid et al., 2012*; *Wessel et al., 2016*; *Wessel et al., 2013b*), we predicted that such motor inhibition – if it indeed takes place during response-conflict – would be broad and non-selective, because it has to be rapidly initiated. In other words, the excitability of even task-unrelated muscles should be suppressed during conflict (as is the case during outright motor stopping). Experiments 3.1. and 3.2. were designed to test this prediction.

Experiment 3.1. was a pilot experiment designed to test the latency with respect to stimulus-onset at which there is non-selective inhibition of the motor system (if there is any). We tested this in N = 14 healthy adults who performed the same Simon task as in Experiment 1, except that in this

version, participants responded with the feet. This allowed us to measure cortico-spinal excitability (CSE) from a hand-muscle (right abductor pollicis brevis, APB) that was not involved in the task. We recorded EMG from APB while stimulating its contralateral motor cortex representation using single-pulses of TMS. Doing so elicits motor-evoked potentials in the EMG trace, a net measurement of CSE (*Bestmann and Duque, 2016*; *Rossini et al., 1994*). We collected CSE-estimates on both incongruent and congruent trials, at different time points following stimulus onset (150 ms – 500 ms post-stimulus in 50 ms increments; roughly covering the post-stimulus period during which electrophysiological signatures purportedly reflecting inhibition were active in Experiments 1, 2).

Participants showed the expected behavior. Correct incongruent trial reaction times were slower compared to congruent trial reaction times (559 ms [SEM: 15] vs. 528 ms [SEM: 15]; t(13)=4.57, p=0.00052, d = 0.56), and error rates on incongruent trials were significantly increased compared to congruent trials (7.04% vs. 5.17%; t(13)=3.56, p=0.0035, d = 0.27), indicating the presence of response-conflict.

Regarding CSE, a repeated-measures ANOVA revealed a significant interaction effect of TRIAL TYPE and TIME POINT on APB-CSE (F(3/39)=3.1, p=0.039, partial-$\eta^2$ =.19). Planned follow-up t-tests showed a significant suppression of the APB-CSE on incongruent relative to congruent trials specifically at 250 ms following stimulus onset (t(39)=2.36, p=0.012, d = 0.52, *Figure 6A*), accounting for the significant interaction. This was significant when corrected for multiple comparisons using the Bonferroni procedure. No other time points showed any significant differences between conditions.

## Experiment 3.2.: Response-conflict leads to non-selective CSE-suppression

We then repeated the experiment with a larger sample (N = 30), while collecting CSE only at the 250 ms post-stimulus time-point that yielded a significant difference in Experiment 3.1. This increased the signal-to-noise ratio for each individual subject by increasing the trial number for the CSE estimate. It also allowed us to correlate CSE suppression to behavioral motor slowing during response-conflict across subjects. Finally, we also collected baseline CSE samples in the breaks between blocks in Experiment 3.2., with the goal of testing whether incongruent-trial CSE was suppressed compared to a baseline that included a true rest-period.

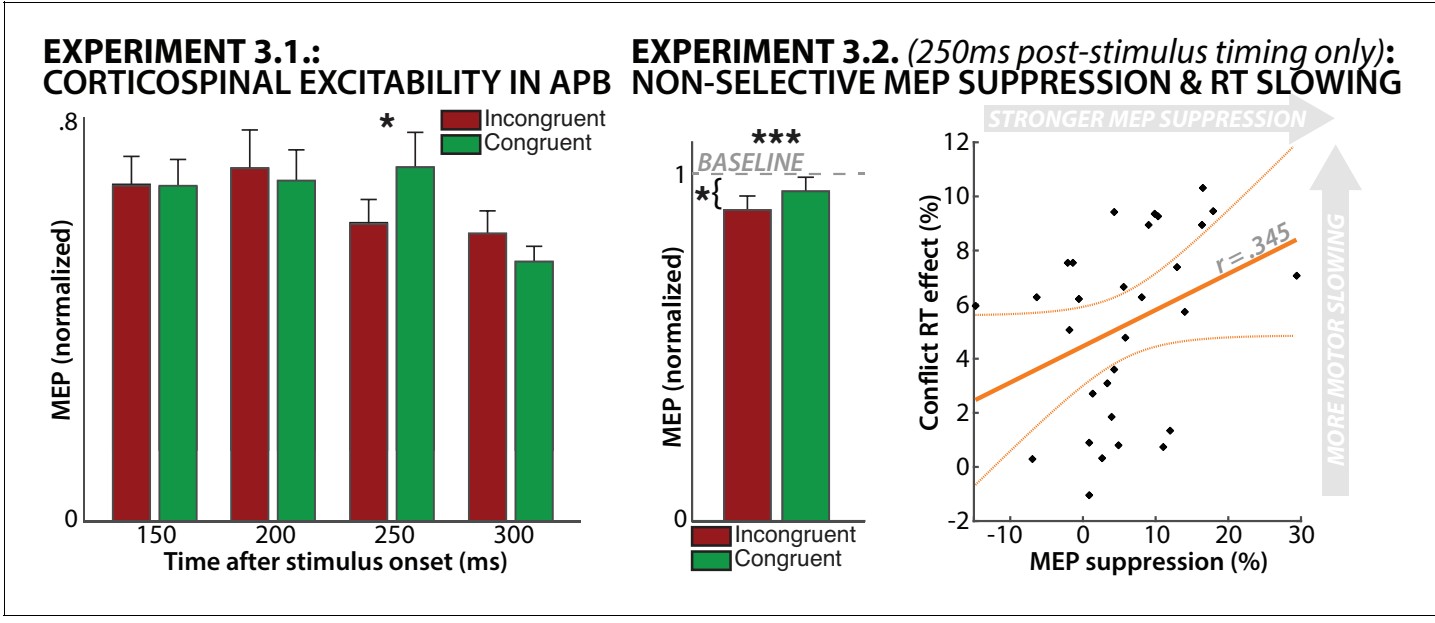

**Figure 6.** Results from Experiment 3.1. (left) and Experiment 3.2. (right). Experiment 3.1. showed significant suppression of CSE on incongruent-trials at 250 ms post-stimulus. Experiment 3.2. replicated this finding in a larger sample, and also showed that incongruent-trial CSE was significantly suppressed compared to resting baseline. Lastly, the degree of MEP suppression incurred on incongruent trials was positively correlated with the amount of relative RT slowing on those trials.
DOI: https://doi.org/10.7554/eLife.42959.009

Participants showed the expected behavior. Correct incongruent trial reaction times were slower compared to congruent trial reaction times (598 ms [SEM: 14] vs. 569 ms [SEM: 15]; t(29)=8.74, p=1.27*10$^{-09}$, d = 0.37), and error rates on incongruent trials were significantly increased compared to congruent trials (4.58% vs. 3.28%; t(29)=2.5, p=0.0185, d = 0.26), indicating the presence of response-conflict.

Just as in Experiment 3.1., CSE at 250 ms post-stimulus was suppressed on incongruent vs. congruent trials (t(29)=3.89, p=0.0005, d=0.24, *Figure 6B*). Moreover, incongruent-trial CSE was suppressed below baseline (t(29)=2.56, p=0.016, d=0.67), while congruent-trial CSE was not (t(29)=1.2, p=0.23). Finally, the amount of RT slowing incurred on incongruent (compared to congruent) trials and the amount of CSE suppression on incongruent (compared to congruent) trials were positively correlated (r = 0.345, p=0.031), though this was only significant when tested in one-sided fashion.

We interpret this finding as a non-selective suppression of motor excitability during response-conflict. However, a potential alternative interpretation is that since feet are an unusual motor effector in the context of a computerized task, the suppressed CSE on incongruent-trials could still reflect selective suppression of the non-responding hand, because an automatic motor tendency may have been co-activated by the task stimuli. To rule out this explanation, we split the trials in Experiment 3.2. by response-side. If the CSE suppression reflects selective suppression of the inappropriate motor tendency at the hand of the non-responding side, we would expect to find a laterality effect, since CSE was only quantified at the right hand. Therefore, we conducted a repeated-measures ANOVA with the factors CONGRUENCY (incongruent, congruent) and RESPONSE SIDE (left, right) to test whether the main effect of CONGRUENCY was moderated by a significant interaction. The main effect of congruency was indeed significant (F(1,29) = 20.5, p=0.00009, partial-$\eta^2$ =.41), in line with the above results from the simple paired-samples t-test. Importantly, the interaction was not significant (F(1,29) = 1.13, p=0.3, partial-$\eta^2$ =.038), showing that RESPONSE SIDE did not influence this main effect. Post-hoc power analyses indicated that this analysis had a substantial amount of achieved power (>0.88). Therefore, we reject the alternative explanation of a laterality effect resulting from selective suppression of a purported motor tendency found at the hand muscle on the non-responding side.

The results of Experiments 3.1. and 3.2. show that CSE is indeed non-selectively suppressed during response-conflict, even in task-unrelated muscles. This suggests that broad, non-selective net-inhibition of the motor system is taking place during response-conflict.

## Discussion

We present data from three experiments investigating the interaction between the motor system and a purported fronto-BG mechanism for inhibitory control during response-conflict. Experiment 1 showed that outright action-stopping and conflict-related response-time slowing relate to similar low-frequency fronto-central scalp-signatures (which may share a common neural generator). This indicates the existence of a common set of cognitive control processes that are triggered in both situations (*Cavanagh and Frank, 2014*). Because of the relationship between this neural activity and the trial-to-trial effects on behavior across tasks, we hypothesized that the one common component is motor inhibition. In line with this, Experiment 2, using intraoperative recordings in PD patients, showed that the STN (the key subcortical node within the purported fronto-BG mechanism for motor inhibition) influences the primary motor cortex representation underlying the incorrect response-tendency. This provides a potential mechanistic explanation for how motor inhibition is enacted during conflict. In support of the inhibitory interpretation of this activity, we also found that increase in STN->M1 communication to be specific to the beta frequency band. STN activity in that frequency band is prominently observed during action-stopping (*Benis et al., 2014*; *Benis et al., 2016*; *Kühn et al., 2004*; *Ray et al., 2012*) and directly predicts the degree of the non-selective suppression of the motor cortex when actions are successfully stopped (*Wessel et al., 2016*). Based on the findings from Experiment 2, combined with this prior knowledge about STN-mediated motor inhibition during outright action-stopping, we therefore expected to find non-selective suppression of CSE during response-conflict. Experiments 3.1. and 3.2. clearly show that CSE is indeed non-selectively suppressed during response-conflict.

Strikingly, despite the fact that all three experiments include fundamentally different types of data, the timing of the activity purportedly indicating inhibition is highly overlapping. Specifically, in

Experiment 1, fronto-central low-frequency scalp-activity was found to start around 200 ms after the onset of incongruent stimuli (*Figure 2C and D*). Accordingly, in Experiment 2, increased STN->M1 connectivity began around the same time (*Figure 5*). Finally, in Experiments 3.1. and 3.2., non-selective suppression of the motor cortex was observed specifically at 250 ms post-stimulus (*Figure 6*). Taken together, this suggests a cascade of processes – indicative of the coordinated effort subserved by a distributed neural network – which ultimately results in the non-selective net-inhibition of incorrect motoric tendencies during response-conflict.

We therefore propose that a fronto-BG mechanism for motor inhibition is recruited as part of cognitive control during response-conflict. We suggest that inhibitory control is a specific mechanism recruited to buttress action-selection by non-selectively suppressing competing, incompatible response tendencies that could potentially lead to erroneous responses. This is in line with several prominent computational models of the basal ganglia, which suggest that this subcortical circuit supports action-selection by actively inhibiting competing motor representations (*Berthet et al., 2012*; *Frank, 2006*; *Gurney et al., 2001*). Our current findings extend these models by showing that the inhibitory influence of the STN is not constrained to the basal ganglia themselves (*Humphries et al., 2006*), but can manifest as a measurable reduction of net-excitability of entire motor tracts during periods of high response-conflict (as indicated by non-selective suppression of CSE). Moreover, our results lend support to the notion that inhibition of motor activity via the fronto-BG network may be a highly generic neural mechanism that can be dynamically deployed across several different control situations (*Wessel and Aron, 2017*). Moreover, they suggest that fronto-BG-mediated motor inhibition can not only be recruited to cancel actions entirely, but can also act as a more fine-grained 'brake' on behavior, to adaptively and gradually slow down movement when cognitive control is needed (*Frank, 2006*; *Wessel, 2018b*).

Classic theories of motor control propose three conceptual BG motor pathways (*Alexander and Crutcher, 1990*; *Duque et al., 2017*; *Graybiel, 2005*; *Groenewegen, 2003*; *Jahanshahi et al., 2015*; *Nambu et al., 2002*). In addition to a pro-kinetic direct pathway (*Hauber, 1998*), two additional anti-kinetic pathways purportedly *inhibit* motor activity: the indirect pathway (*Parent and Hazrati, 1993*; *Parent and Hazrati, 1995*; *Smith et al., 1998*) and the hyper-direct pathway (*Nambu et al., 2002*). Both inhibitory pathways feature the STN and net-inhibit thalamo-cortical motor representations via the output nuclei of the BG. The hyper-direct pathway purportedly skips the (external) pallidal and striatal parts of the indirect pathway and is thought to exert very rapid net-inhibition (*Inase et al., 1999*; *Kelley et al., 2018*; *Nambu et al., 2002*). Some studies suggest that BG circuitry can indeed be recruited in different inhibitory 'modes'. Specifically, when rapid, reactive control is needed, it is thought to involve the hyper-direct pathway (*Aron and Poldrack, 2006*), whereas when control can be proactively engaged (for example, when the to-be-inhibited effector is known prior to motor initiation), motor inhibition involves the striatum and pallidum, which is more consistent with the indirect pathway (*Majid et al., 2013*). Furthermore, it has been suggested that non-selective CSE-suppression results from rapid, reactive recruitment of the STN, purportedly via the hyper-direct pathway (*Duque et al., 2017*; *Wessel et al., 2016*). Indeed, non-selective CSE suppression cannot be found during proactive motor inhibition, which purportedly involves the indirect pathway (*Greenhouse et al., 2012*). Hence, it is possible that response-conflict is resolved via a hyper-direct pathway, which is rapidly recruited to suppress the conflicting response tendency. In doing so, it temporarily suppresses the motor system in a non-selective fashion. However, to truly distinguish between the indirect and hyper-direct pathway contributions to inhibition during response-conflict, further studies are needed.

It remains an open question whether this broad, non-selective suppression of motor excitability is a 'side-effect' of very rapidly exerted inhibition, or whether it is functionally beneficial. A non-selective, broad 'brake' on behavior, which rapidly and non-selectively increases the global motor threshold during situations like response-conflict, may indeed be beneficial to behavior – akin to the proposed 'hold your horses' signal implemented via the STN during decision-conflict (*Cavanagh et al., 2011*; *Frank, 2006*; *Frank et al., 2007*; *Herz et al., 2017*; *Herz et al., 2016*). A potential benefit could be to buy time for slower, more precise cognitive control mechanisms that enact more fine-grained control. During motoric response-conflict, it makes sense to assume that a rapid (rather than a slower) inhibitory system would be initially recruited to brake movement, as responses in tasks like the Simon task are made under time pressure and may not allow for fine-grained control.

Our results suggest a broader role for motor inhibition processes in cognitive control. In a recent theoretical account (*Wessel and Aron, 2017*), we proposed that motor inhibition could be a ubiquitous, universal mechanism engaged by surprising events of different types (action errors, unexpected action outcomes, unexpected perceptual events). In fact, surprising events may be exemplars of an even broader category of control situations – namely, those that involve uncertainty. Uncertainty could be a common principle that includes environmental volatility (e.g., unexpected events; *Wessel, 2018b*), perceptual ambiguity (*Herz et al., 2017*; *Herz et al., 2016*), value-based decision-conflict (*Cavanagh et al., 2011*; *Frank, 2006*), and motor uncertainty (e.g., response-conflict). According to *Frank (2006)*, uncertainty (in their case, during value-based decision-making) invokes a 'global' No-Go command issued by the STN. In line with this, our results suggest a concrete physiological mechanism by which such 'global' No-Go commands could inhibit the motor system during action-selection under uncertainty. An open question is conflict that arises from competing, simultaneous activations of several competing motor representations (as in the Simon task) involves the same mechanisms that are active when conflict results from perceptual ambiguity (e.g., in moving-dot paradigms [*Brittain et al., 2012*; *Herz et al., 2017*; *Herz et al., 2016*]) or decision-related ambiguity (e.g., during value-based decision making [*Cavanagh et al., 2011*; *Frank et al., 2007*]).

There are some limitations to the current study. First, while our study provides converging evidence from several imaging domains, it provides no causal evidence for the involvement of fronto-BG circuitry for motor inhibition during response-conflict. However, it provides a potential target for future causal investigations. For example, direct-electrical stimulation of the fronto-BG network leads to a momentary slowing of motor execution when action-stopping is anticipated in the SST (*Wessel et al., 2013a*). Such stimulation during response-conflict could lead to similar effects. Indeed, a single-patient study found increased conflict-related motor slowing during direct-electrical stimulation of the mPFC (*Usami et al., 2013*), supporting the idea that the associated network momentarily inhibits behavior. Our current study would lead to the prediction that such stimulation of cortical nodes of the fronto-BG network should lead to increased non-selective CSE suppression. Second, our study does not offer any hints as to the generative models or biophysiology underlying the low-frequency and β-rhythms that are functionally implicated in our experiments. Specifically, while the finding that conflict and action-stopping evoke low-frequency activity in mPFC, whereas the BG and the motor-system communicate in the β-band, is in line with the prior literature, no study so far offers any mechanism by which low-frequency interactions between mPFC and STN are translated into β-frequency interactions between the BG and M1. Interestingly, a recent study has found that low-frequency and β-band activity may influence conflict-adaptation at different times of the trial, and that cross-regional increases in synchrony at low frequencies are independent of the frequency of local activity in STN and prefrontal cortex during these adaptive behaviors (*Zavala et al., 2018*).

Lastly, while we prefer to interpret our findings in the context of motor inhibition, incongruent and congruent Simon-task trials notably differ in other aspects, such as attentional processes, difficulty, and arousal. However, we argue that none of these factors can account for the entire observed pattern of data, across all four experiments, in the same way that inhibitory control can. An arousal-based explanation is at odds with CSE-suppression during conflict, since phasic arousal leads to increases in CSE (*Baumgartner et al., 2007*; *Hajcak et al., 2007*). A difficulty-based explanation is at odds with the fact that such CSE suppression is consistently greater on successful vs. failed stop-trials in the SST (*Badry et al., 2009*; *Cai et al., 2012*; *Majid et al., 2012*; *Wessel et al., 2013b*), even though failed stop-trials were more difficult. Furthermore, it is unclear why either an arousal or difficulty-related brain process would show the timing differences exhibited by the EEG signal in Experiment 1 (earlier onset on successful vs. failed stop-trials in the absence of amplitude differences). The race-model of motor inhibition, however, predicts exactly that pattern. An attention-related explanation is hardest to rule out, given that attention and motor inhibition are tightly (and maybe inextricably) linked (*Wessel, 2018b*; *Wessel and Aron, 2017*). However, the suppression of motor excitability, as well as increased communication between the BG and M1 are physiologically and theoretically coherent with inhibition, while few (if any accounts) exist that relate these signatures to attention.

One notable aspect about our study is the absence of connectivity between STN and *contra*lateral M1 during response-conflict in Experiment 2. Experiments 1 and 3 clearly show that there is an

association between inhibitory signatures at scalp- and motor-system level and the degree of conflict-related behavioral slowing observed at the correct effector. This begs the question of why the increase in coherence between STN and M1 is only found at the ipsilateral side, whereas no comparable coherence between the M1 representation of that effector and the STN could be observed. (However, this null-finding has to be interpreted with caution, given the fact that Bayes Factor analyses only show moderate positive evidence for the null hypothesis; cf. *Figure 5—figure supplement 1*). One possible explanation is that even when conflict is detected, the motor representation of the correct response in contralateral M1 must still be subject to ongoing *excitation* by the direct pathway (unlike the incorrect response tendency), since that response still has to be executed. Notably, the competition between the inhibitory process incurred by the conflict and the excitatory process incurred by the goal to rapidly execute the correct response may be negotiated at levels of the motor system that do not involve M1 (e.g., in interactions between the STN, pallidum, striatum, and/or thalamus, cf. *Schmidt et al., 2013*). Another alternative explanation is that the simultaneous excitation and inhibition of contralateral M1 during response-conflict could have mutually countermanding effects on M1, which could obscure the degree to which inhibitory influence of STN on M1 is measurable using the connectivity techniques used here. Future studies that simultaneously image several basal ganglia nuclei may be able to resolve this question.

In summary, we report converging evidence for the involvement of a fronto-BG inhibitory control mechanism during motoric response-conflict. Across modalities, the signatures of this mechanism were directly related to conflict-related slowing of motor execution. We therefore propose that this mechanism reflects an inhibitory effort aimed at rapidly suppressing inappropriate motor tendencies, resulting in non-selective suppression of CSE, which could be part of a wider cascade of common, universally deployable control processes (*Cavanagh and Frank, 2014*; *Wessel and Aron, 2017*). Our data provide novel insights into the specific neural mechanisms by which inhibition is exerted during response-conflict, and suggest a theory of BG function according to which a low-latency inhibitory pathway can rapidly and non-selectively suppress cortico-motor activity during action-selection under conflict.

## Materials and methods

### Experimental model and subject details

In Experiment 1, N = 22 healthy adult human volunteers were recruited from the wider University of Iowa community via an emailing list for research subjects and were paid an hourly fee of $15. One dataset had to be excluded due to a technical problem with the EEG recording hardware, leaving a final sample of N = 21 (mean age: 27.4y, SEM: 2.24; 11 female, 10 male). This sample size was chosen to match our prior report of independent component-based single-trial EEG activity across two separate cognitive tasks (*Wessel, 2018a*). Informed consent was collected from all subjects and all procedures were approved by the local ethics committee at the University of Iowa (IRB #201511709).

For Experiment 2, patients were recruited opportunistically from all available STN neurosurgical candidates in the University of Iowa Neurosurgery Clinic over a time period of three years. Given the limited sample size, we verified our group-level results on the single-subject level where possible (cf. *Figure 5*). One dataset had to be excluded because of chance-level performance (error rate on incongruent trials: 75.9%), leaving a final sample of N = 9 (mean age: 60.8, SEM: 3.04). Informed consent was collected from all subjects and all procedures were approved by the local ethics committee at the University of Iowa (IRB # 201402720).

In Experiment 3.1., N = 15 healthy adult human volunteers completed the experiment. Since this was a pilot study, no a priori sample size computation was performed. Subjects were recruited from the wider University of Iowa community via an emailing list for research subjects and were paid an hourly fee of $15. One dataset had to be excluded due to artifact contamination of the EMG trace, leaving a sample of N = 14 (mean age: 24.8y, SEM: 1.4; seven female, seven male). Informed consent was collected from all subjects and all procedures were approved by the local ethics committee at the University of Iowa (IRB #201612707).

In Experiment 3.2., N = 30 healthy adult human volunteers (mean age: 19.4y, SEM:. 4; 19 female, 11 male) completed the experiment. This sample size was chosen based on the results of the pilot

study, which indicated that based on the effect size for the 250 ms time point, 30 participants are necessary to achieve a two-sided power of .9 to detect a CSE suppression effect. Subjects were recruited via the research subject recruitment tool in the Department of Psychological and Brain Sciences at the University of Iowa and received course credit in exchange for participation. Informed consent was collected from all subjects and all procedures were approved by the local ethics committee at the University of Iowa (IRB #201612707).

## Method details

### Experimental paradigms

#### Visual Simon task (Experiments 1, 3.1., 3.2.)

All task code for Psychtoolbox version 3 (*Brainard, 1997*) can be downloaded on the Open Science Framework at the following URL: https://osf.io/k3ypt/.Participants responded to the color of a square stimulus presented to either side of a central fixation cross. The side of stimulus presentation was irrelevant to the response. Each trial consisted of a fixation screen consisting of a central fixation cross and the button mappings presented on the bottom of the screen (duration: 500 ms), followed by stimulus presentation (200 ms), followed by the response, followed by an inter-trial interval (overall trial duration fixed at 3,500 ms). Participants responded with their left hand to the colors blue and cyan ('cold' colors) and with their right hand to the colors yellow and red ('warm' colors). Responses were made by pressing the q or p keys on a QWERTY keyboard (Experiment 1) or by pressing down on USB foot pedals (Kinesis Savant Elite 2) with the left/right foot (Experiments 3.1., 3.2.). Participants performed eight total blocks of 96 trials each (48 congruent, 48 incongruent).

#### Auditory Simon task (Experiment 2)

Participants responded to the pitch of a 200 ms sine-wave tone presented either through the left or right side of an in-ear headphone. As in the visual task, the side of stimulus presentation was irrelevant to the response. Participants were instructed to respond on USB pedals (Kinesis Savant Elite 2) using their left thumb following a 500 Hz tone and using their right thumb following a 1,200 Hz tone. After stimulus presentation, participants had 2,000 ms to respond. After the response, a 1,500 ms inter-trial interval begin. Each block of the experiment consisted of 80 trials (40 congruent, 40 incongruent).

#### Stop-signal task (Experiment 1)

Trials began with a white fixation cross on a gray background (500 ms duration), followed by a white leftward or rightward arrow (go-signal). Participants had to respond as fast and accurately as possible to the arrow using their left and right index finger (the respective response-buttons were q and p on the QWERTY keyboard). On 33% of trials, a stop-signal occurred (the arrow turned from white to red) at a delay after the go-stimulus (stop-signal delay, SSD). The SSD, which was initially set to 200 ms, was dynamically adjusted in 50 ms increments to achieve a p(stop) of. 5: after successful stops, the SSD was prolonged; after failed stops, it was shortened. This was done independently for leftward and rightward go-stimuli: SSD started at 200 ms for both left- and right-arrow trials. Then, if a stop-trial with a leftward arrow lead to a failed stop, the SSD for the next leftward arrow was shortened by 50 ms, whereas the SSD for the next rightward response remained unchanged. This way, the SSD was allowed to vary independently for each arrow/response direction. Trial duration was fixed at 3000 ms. Six blocks of 50 trials were performed (200 go, 100 stop).

### Electrophysiological recordings

#### Extracranial EEG recording (Experiment 1)

EEG was recorded using a 62-channel electrode cap connected to two BrainVision MRplus amplifiers (BrainProducts, Garching, Germany). Two additional electrodes were placed on the left canthus (over the lateral part of the orbital bone of the left eye) and over the part of the orbital bone directly below the left eye. The ground was placed at electrode Fz, and the reference was placed at electrode Pz. EEG was digitized at a sampling rate of 500 Hz, with hardware filters set to 10 s time-constant high-pass and 1000 Hz low-pass.

### Local field potential recordings (Experiment 2)

Intrasurgical recordings were made from STN via four macroelectrode contacts on each side DBS lead (3387, Medtronic, Inc, Minneapolis, MN) and from two four-contact subgaleal electrode strips (M1; Ad-Tech, Oak Creek, WI; 10 mm spacing center-to-center, 3 mm exposed contact diameter). The strip electrodes were inserted just posterior to the surgical burr hole (located at the coronal suture) in a para-sagittal (caudal) direction and anterior-posterior alignment, in an effort to cover the precentral gyrus (see below for additional electrode positioning detail). Recordings were made using a Tucker-Davis Technologies (Alachua, FL) recording system using a RA16PA 16-Channel Medusa pre-amplifier and a RA16LI head-stage at a sampling rate of 24,414 Hz, with a hardware-side low-pass filter of 7.5 kHz (gain: $10^6$). Line noise artifacts introduced by the OR environment were removed from the raw data trace using the CleanLine plug-in for EEGLAB (https://www.nitrc.org/projects/cleanline/). Data were manually checked for time periods with artifacts, which were removed prior to analysis.

### EMG recording (Experiments 3.1., 3.2.)

EMG recording methods and equipment were identical to *Dutra et al. (2018)*. EMG was recorded using a bipolar belly-tendon montage over the first dorsal interosseous muscle (FDI) of the right hand using adhesive electrodes (H124SG, Covidien Ltd., Dublin, Ireland), with a ground electrode placed over distal end of ulna. Electrodes were connected to a Grass P511 amplifier (Grass Products, West Warwick, RI; 1000 Hz sampling rate, filters: 30 Hz high-pass, 1000 Hz low-pass, 60 Hz notch). The amplified EMG data were sampled via a CED Micro 1401–3 sampler (Cambridge Electronic Design Ltd., Cambridge, UK) and recorded to the disc using CED Signal software (Version 6).

### TMS stimulation (Experiments 3.1., 3.2.)

TMS stimulation methods and equipment were identical to *Dutra et al. (2018)*. Cortico-spinal excitability (CSE) was measured via motor-evoked potentials (MEPs) elicited by TMS. TMS stimulation was performed with a MagStim 200–2 system (MagStim, Whitland, UK) with a 70 mm figure-of-eight coil. Hotspotting was performed to identify the FDI stimulation locus and correct intensity. The coil was first placed 5 cm lateral and 2 cm anterior to the vertex and repositioned to where the largest MEPs were observed consistently. Resting motor threshold (RMT) was then defined as the minimum intensity required to induce MEPs of amplitudes exceeding. 1 mV peak to peak in 5 of 10 consecutive probes (*Rossini et al., 1994*). TMS stimulation intensity was then adjusted to 115% of RMT (Experiment 3.1.: mean intensity: 48.9% of maximum stimulator output; range: 37–54%; Experiment 3.2.: mean intensity: 62.5% of maximum stimulator output; range: 46–79%) for stimulation during the experimental task. In Experiment 3.1., TMS pulses were timed to occur at one of eight time points after Simon-stimulus onset (evenly spaced in 50 ms increments ranging from 150 to 500 ms). In case a response was made before the TMS pulse, no pulse was triggered. An EMG sweep was triggered 150 ms before each TMS pulse.

To normalize MEP amplitudes between subjects, we also collected five baseline samples per block. These were randomly interspersed between the trials and occurred during 'null' trials, in which no imperative stimulus followed the fixation screen. Therefore, in Experiment 3.1., we collected 40 congruent and 40 incongruent trials per time-point across all blocks, plus 40 baseline trials.

In Experiment 3.2., TMS pulses only occurred at 250 ms after task-stimulus onset. Additionally, in that experiment, we also collected estimates for a true resting baseline; we collected 10 trials of MEP during each break between blocks, as well as immediately before and after the experiment.

## Procedural overviews

### Procedure experiment 1

After signed written informed-consent, EEG caps were affixed to the participants' scalp. Participants then practiced and performed the Simon task, followed by the stop-signal task. The order of tasks was fixed to avoid potentially biasing participants towards using inhibitory control in the Simon task.

### Procedure experiment 2

Participants signed written informed-consent during a pre-surgical visit to the neurosurgery clinic. If possible, we performed a first practice session with the auditory Simon task during the pre-surgery

visit. On the day of the surgery, participants practiced the task before the surgery was initiated, until performance indicated that they had understood and could successfully perform the task. After the implantation of the first DBS lead (and both subgaleal strip electrodes), during the time at which the patients had to be awake to perform neurological assessments to confirm the placement of the DBS lead, they performed one block of the auditory Simon task with congruent trials only (40 trials). This was done to confirm adequate position of the subgaleal strip electrodes. In all patients we utilized four-contact strip electrodes with 10 mm center-to-center inter-electrode spacing, which were placed into the subgaleal space over bilateral motor cortex through the surgical incision for the burr hole (no additional incision was made to place these leads). The burr holes were located at the coronal suture and 4.1 cm lateral to the midline on average (range 3.3–4.9 cm, N = 6 patients with measurements noted in medical record). We have previously utilized similar methodology for subdural positioning over motor cortex with the same style of strip electrodes (*Kelley et al., 2018*). For the current study, we estimate the strip electrodes to be situated approximately 10 mm more posterior than described in the 2018 study since the current subgaleal strips did not have to go through the burr hole; therefore the most posterior electrode in the current series is estimated to be ~6 cm posterior to the coronal suture, consistent with sufficient posterior placement to cover precentral gyrus/ motor cortex (*Park et al., 2007*; *Rivet et al., 2004*). We then analyzed the recordings from the subgaleal motor electrodes while in the operating room. If no visible motor signature (i.e., β-suppression time-locked to the response following the congruent Simon stimuli) was found, the lead placement on the respective side was changed after the clinical assessment was completed. While we did not repeat this functional motor localization procedure after the subgaleal lead placement was changed (to avoid elongating the surgical procedure), this did give us one chance to try one different placement before the actual experiment began if the initial placement did not capture motor activity. In the one patient in which the original placement did not yield a reliable motor signal, we confirmed its presence from the congruent trial data of the actual experiment data itself. After implantation of the second DBS lead and clinical validation of its placement, participants performed the full version of the experiment. Five participants performed four blocks (320 total trials), one participant performed five blocks (400 trials), one participants performed three blocks (240 trials), one performed two blocks (120 trials), and one performed one block (80 trials).

## Procedure experiments 3.1. and 3.2

After signing written-informed consent, participants underwent a TMS safety questionnaire (*Rossi et al., 2011*). After that, the hotspotting procedure began (see above). After the appropriate hotspot and stimulation intensity were identified, the experiment began.

## Data preprocessing

### Extracranial EEG preprocessing (Experiment 1)

Data were preprocessed using custom routines in MATLAB. ICA was performed using functions from the EEGLAB toolbox (*Delorme and Makeig, 2004*). After import into MATLAB, data from both tasks were merged, and the continuous time-series were filtered using symmetric 2-way least-squares finite impulse response filters with a high-pass cutoff of 0.5 Hz and a low-pass cutoff of 50 Hz. The continuous time-series were then visually inspected for channels with non-stereotypic artifacts, which were excluded from further processing. The remaining data were visually inspected for segments with non-stereotyped artifact activity (e.g., muscle artifacts), which were removed from further analysis of the continuous data. After artifact removal, the data were re-referenced to the common average, and subjected to a temporal infomax ICA decomposition algorithm (*Bell and Sejnowski, 1995*), with extension to subgaussian sources (*Lee et al., 1999*). The resulting component matrix was screened for components representing eye-movement artifacts using outlier statistics. The IC selection was visually inspected for accuracy of the automated classification, and additional artifact components were removed. The remaining components were subjected to further analyses.

### EEG analysis: independent component selection (Experiment 1)

We selected a single independent component from each participant's ICA transformation of the entire EEG dataset. This component was selected to reflect the properties of a neural motor inhibition process in the stop-signal task. This was done according to the procedure first introduced in

one of our prior studies (*Wessel and Aron, 2015*), and further described in results section and below.

To identify these components in an algorithmic, quantifiable fashion, we used a procedure that is based on the COMPASS algorithm (*Wessel and Ullsperger, 2011*). From each individual participant's ICA, we first selected each component whose weight matrix had its maximal weight at one of the frontocentral electrodes (FCz, Cz, C1, C2, FC1, FC2). We then averaged those components' back-projected channel-space activity at these fronto-central electrodes within the 500 ms time-period following the stop-signal, and correlated this event-related average activity to the event-related average activity of the overall EEG data (i.e., the EEG data based on the back-projection of *all* non-artifact ICs for that participant) in that time range. The component that showed the highest correlation with the overall ERP was selected as the motor inhibition component (as in the 500 ms post-stop-signal time period, the fronto-central ERP will be dominated by the inhibition-related activity). In two cases, more than one component explained significant parts of the channel-space P3. In these two cases, both components were included in the analyses (we also performed the same analyses with those two subjects excluded from the sample, with virtually identical results).

To test whether the thusly selected components from our current study showed the same motor inhibition-related properties as described in previous studies, we detected the onset of each participant's fronto-central P3 back-projection based on the selected component. Again, this was done in accordance with our prior studies (introduced in *Wessel and Aron, 2015*): For each subject, we selected four groups of trials – successful stop-trials, failed stop-trials, and a set of matched go-trials for each type of stop-trial. Specifically, 'matched' go-trials were selected by using a subset of go-trials on which the SSD staircase was at the same position as the trials included in the successful/failed stop-trial sample. Then, the difference between successful/failed stop-trials and the respective sample of matched go-trials was tested for differences from zero using sample-by-sample paired t-tests between the trial-to-trial amplitudes at a two-sided p=0.01. These t-tests were performed on each sample in the time period ranging from 0 to 500 ms following the stop-signal (with the individual time points in the stop-signal waveforms being compared to matched time points in the go-trial waveforms – that is, the timepoints in the go-trial waveform at which the stop-signal would have appeared according to the current SSD on that trial). This resulted in a vector of 250 logical values that showed at which sample points there was a significant difference between successful/failed stop- and matched go-trials. To identify the exact onset of the P3, the peak of the P3 difference-wave in the critical time period (0–500 ms following the stop-signal) was then detected. The t-test at that peak sample was significantly different from 0 in all cases – that is, all subjects showed a significant P3 for both successful- and failed stop-trials at least at the peak sample. Working 'backwards' from that peak sample, we then identified the sample closest to the onset of the stop-signal at which the positive difference wave was still significantly different from zero (at p<0.01). Put differently, we defined the P3 onset in each participant as the time point after the stop-signal at which the stretch of significant samples that included the peak of the P3 began. Again, this was done separately for successful and failed stop-trials. We then compared the thusly-identified single-subject onsets between successful and failed stop-trials using a paired-samples t-test to test the relationship between stopping-success and the onset timing of these components within participants. Furthermore, the onset timing of this component on successful stop-trials was also correlated to SSRT to test the relationship between stopping efficacy and the onset timing of the selected P3 component between participants. This procedure is identical to our initial report of the relationship between P3 onset and SSRT/stopping success (*Wessel and Aron, 2015*).

## STN and subgaleal motor strip electrode selection (Experiment 2)

Subthalamic nucleus activity was recorded bilaterally using four macro-electrode contacts per side. For each side, three bipolar reference montages were generated by means of subtraction of immediately adjacent contacts. We then selected the bipolar montage with the clearest β-band peak during resting activity (*Figure 3B*) and subsequently averaged the activity from left and right STN for each subject.

Motor activity was recorded from bilateral M1 using subgaleal electrode strips. Just like for the STN, for each side, three bipolar reference montages were generated by means of subtraction of immediately adjacent contacts. We selected the montage with the clearest motor signature on each side (β-desynchronization in the peri-operative localization experiment, *Figure 4*) for further

analyses. For the analyses of the ipsilateral M1 connectivity with STN, 'ipsilateral' M1 activity was defined as the activity from the selected M1 contact in the hemisphere ipsilateral to the correct response hand. For visualization purposes, we also quantified the β-desynchronization at immediately adjacent contacts on the four-electrode strip. That is, for subjects in which the middle pair of contacts was selected, we also plotted the outer (anterior and posterior) montages, whereas for subjects in which an outer pair of contacts was selected, we also plotted the immediately adjacent pair of contacts. We then created two comparisons of subjects, one subset of subjects who had contacts anterior to the selected contacts, and one subset of subjects that had contacts posterior to the selected contacts. We then plotted that activity to investigate whether there were local gradients in the purported β-desynchronization that would indicate whether the activity was localized (*Figure 4*).

## Time-frequency power (Experiment 1)

To compute time-frequency power, we used a filter-hilbert method. In short, the whole time-series (EEG channel-data/LFP trace) was filtered at 30 center-frequencies (integers ranging from 1 to 30 Hz), with a frequency window of +/-. 5 Hz using symmetric 2-way least-squares finite impulse response filters. The filtered time-series were then translated into complex space using the Hilbert transform (as implemented in the MATLAB hilbert() function). An analytical signal was then extracted by computing the square of the absolute of the complex output of the Hilbert transform.

## Phase-locking value (Experiment 2)

Phase-locking value (PLV) was computed from the phase-angle of the Hilbert transform based on the equation

$$ISPCtf = \left| n^{-1} \sum_{r=1}^{n} e^{(i\theta tf, r, x - i\theta tf, r, y)} \right|$$

where *n* denotes the total number of trials *r*, θ denotes the phase angle, t denotes time, f denotes frequency, and x and y denote the two electrode sites in question. Estimates of PLV for each condition were corrected to a pre-stimulus baseline ranging from 300 to 0 ms.

## Granger prediction analysis (Experiment 2)

To test the directional influence between STN and ipsilateral M1 underlying the increased phase-locking connectivity, we used a Granger-prediction analysis, which allows for a quantification of directional connectivity based on a bivariate autoregressive model (*Granger, 1969*). This analysis was implemented using code adapted from M. X. Cohen (*Cohen, 2014*). Bidirectional (STN to M1, M1 to STN) Granger-prediction estimates were derived in the time window ranging from −300 to 500 ms post-stimulus on incongruent trials, using the same contacts that were underlying the PLV analysis. Estimates were compared to pre-stimulus baseline (0–300 ms) in 21 individual frequencies focusing on the β frequency-range (10–30 Hz) – that is, the frequency-range in which PLV differences between incongruent and congruent trials had been significant. This was done by calculating percent change from baseline (mean Granger-prediction estimates for each frequency in the baseline period subtracted from the post-stimulus estimates, divided by the baseline, and multiplied by 100). We chose a model order of 50 and a window length of 200 ms as a compromise between a) optimal frequency-resolution in the β-band, b) high temporal precision of the estimates, and c) stability of the autoregressive model. Since this trade-off varies between frequencies, leading to different preferable model orders for different frequencies, depending on the desired time/frequency-precision (*Cohen, 2014*), and since we had a strong a priori hypothesis about the β-band (given the results from the PLV analysis), we decided to band-restrict this analysis to the beta-band. This avoids having compare Granger coefficients across frequencies that results from multiple autoregressive models with different model orders.

## EMG/MEP preprocessing (Experiments 3.1., 3.2.)

MEPs were identified from the EMG trace via in-house software developed in MATLAB. Trials were excluded if the root mean square power of the EMG trace 100 ms before the TMS pulse exceeded .01 mV or if the MEP amplitude did not exceed .01 mV. MEP amplitude was quantified with a peak-to-peak rationale, measuring the difference between maximum and minimum amplitude within a

time period of 10–50 ms after the pulse. Both automated artifact rejection and MEP amplitude quantification were visually checked for accuracy on each individual trial for every data set by a rater who was blind to the specific trial type. We then calculated the median MEP amplitudes for each condition of interest (incongruent, congruent), and normalized by dividing these amplitudes by the median baseline MEP estimate.

## Quantification and statistical analysis

### Behavioral data analysis

For the Simon tasks, reaction times and error rates were computed by averaging each condition (incongruent, congruent) separately. Correct-trial reaction times and error rates were compared using paired-samples t-tests.

For the stop-signal tasks, reaction times for correct go- and failed stop-trials were computed and compared using paired-samples t-tests. Mean SSRT and stopping success rates were also computed. SSRT was computed using the integration method.

### EEG data analysis (Experiment 1)

After independent component selection (see above), the activity of the selected component was back-projected into channel space by means of selective matrix multiplication. The signal at fronto-central electrodes FCz and Cz was then averaged, resulting in one time-series per subject that reflected the fronto-central activity of the independent signal component selected from the stop-signal data. We then averaged the activity in this time-series with respect to the events in the Simon task for each subject individually. ERSPs were then computed as described above. Before statistical comparisons were made, the signal was transformed into decibels using 300 ms to 0 ms prior to stimulus onset as the baseline period. Differences between correct congruent and correct incongruent trials were then tested for significance on the group level using sample-to-sample paired-samples t-testing for each time-point in the time-frequency window spanning frequencies from 1 to 30 Hz and time points from 0 to 700 ms following stimulus onset in the Simon task. The resulting 30*350-point matrix of p-values was corrected for multiple comparisons using the false-discovery rate method (*Benjamini et al., 2006*) to a critical p-value of p<0.01.

To test the correlation between correct incongruent trial reaction time and this EEG signal, a trial * time * frequency single-trial matrix of EEG activity was generated for each subject. Each data point in each subjects' time-frequency matrix (30*350) was then modeled using a general linear model that used the standardized (z-scored) reaction time of as a predictor. The resulting subjects * frequencies * time-matrix of beta-weights was tested for significant differences from zero on the group-level using the same sample-to-sample t-test method as described above.

### Local field potential, phase-locking value, and Granger prediction analysis (Experiment 2)

STN-M1 phase-locking value and Granger estimates were computed as described above, with baseline periods of −300 to 0 ms leading up to stimulus onset. Differences between correct congruent and correct incongruent trials were tested for significance on the group level using sample-to-sample t-testing for each time-point in the time-frequency window spanning frequencies from 1 to 30 Hz and time points from 0 to 500 ms following stimulus onset in the Simon task (ERSPs and PLV). Changes in Granger-prediction estimates from pre-stimulus baseline on incongruent trials were tested against 0 using the same method. All p-thresholds for the Granger-analyses were one-sided, as Granger coefficients cannot be negative.

### MEP analysis (Experiments 3.1., 3.2.)

MEPs were averaged separately for each condition (correct incongruent trials and correct congruent trials) and normalized by dividing the mean amplitude by the mean of the baseline MEP samples.

In Experiment 3.1., amplitudes were tested using a 2 × 4 repeated-measures ANOVA with the factors TRIAL TYPE (correct congruent, correct incongruent) and TIME POINT (150, 200, 250, 300 post-stimulus). The later four time points that were collected in the actual experiment (350, 400, 450, and 500 ms post-stimulus) were removed from this analysis, as several subjects had less than 10 trials per condition at at least one of these time points (this was because no TMS pulse was triggered

when a response was made before the pulse was slated to occur, which is increasingly likely at later stimulation time points). Trial numbers for the eight remaining conditions (four individual time points for each congruent/incongruent trial type) ranged from 12 to 36, averaging between 23.36 and 27.5 trials. Follow-up tests were done on individual conditions using paired-samples t-tests.

In Experiment 3.2., amplitudes for correct congruent and correct incongruent trials were tested using a paired-samples t-test.

### Data and software availability

The experimental code, data, and analysis routines underlying this research can be found on the Open Science Framework at the following URL: https://osf.io/k3ypt/.

## Acknowledgements

The authors would like to thank Haiming Chen and Araceli Ramirez-Cardenas for assistance with the intrasurgical recordings; Hailey Billings, Kylie Dolan, Nathan Chalkley, Kristi Hendrickson, and Cailey Parker for help with data collection on the EEG and TMS experiments; our patients for their participation; and Ian Greenhouse and Nicki Swann for helpful comments on the manuscript. This research was supported by grants from the National Institute of Health (R01 NS102201 to JRW and T32 GM108540 to DAW) as well as the Roy J Carver Foundation (JRW).

## Additional information

### Funding

| Funder | Grant reference number | Author |
| --- | --- | --- |
| National Institute of Neurological Disorders and Stroke | R01 NS102201 | Jan R Wessel |
| National Science Foundation | CAREER 1752355 | Jan R Wessel |
| Roy J. Carver Charitable Trust | | Jan R Wessel |
| National Institute of General Medical Sciences | T32 GM108540 | Darcy A Waller |

The funders had no role in study design, data collection and interpretation, or the decision to submit the work for publication.

### Author contributions

Jan R Wessel, Conceptualization, Software, Formal analysis, Supervision, Funding acquisition, Investigation, Visualization, Methodology, Writing—original draft, Project administration, Writing—review and editing; Darcy A Waller, Data curation, Formal analysis, Investigation, Writing—review and editing; Jeremy DW Greenlee, Conceptualization, Investigation, Writing—review and editing

### Author ORCIDs

Jan R Wessel (iD) https://orcid.org/0000-0002-7298-6601
Darcy A Waller (iD) https://orcid.org/0000-0002-3489-7624

### Ethics

Human subjects: Informed consent was collected from all subjects and all procedures were approved by the local ethics committee at the University of Iowa (IRB #201511709, IRB # 201402720, IRB #201612707).

### Decision letter and Author response

Decision letter https://doi.org/10.7554/eLife.42959.014
Author response https://doi.org/10.7554/eLife.42959.015

# Additional files

## Supplementary files

• Transparent reporting form
DOI: https://doi.org/10.7554/eLife.42959.010

## Data availability

The experimental code, data, and analysis routines underlying this research can be found on the Open Science Framework at the following URL: https://osf.io/k3ypt/.

The following dataset was generated:

| Author(s) | Year | Dataset title | Dataset URL | Database and Identifier |
|---|---|---|---|---|
| Wessel JR, Waller DA, Greenlee JDW | 2019 | Data from: Non-selective inhibition of inappropriate motor-tendencies during response-conflict by a fronto-subthalamic mechanism | https://osf.io/k3ypt/ | Open Science Framework, osf.io/k3ypt |

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
