## [Decision Letter]

Thank you for submitting your article "Non-selective inhibition of inappropriate motor-tendencies during response-conflict by a fronto-subthalamic mechanism" for consideration by *eLife*. Your article has been reviewed by three peer reviewers, including Tim Verstynen as the Reviewing Editor and Reviewer #1, and the evaluation has been overseen by Richard Ivry as the Senior Editor. The following individuals involved in review of your submission have agreed to reveal their identity: Zafeirios Fountas (Reviewer #2).

The reviewers have discussed the reviews with one another and Tim has drafted this decision to help you prepare a revised submission.

Summary:

This manuscript by Wessel and colleagues describes a diverse set of experiments designed to characterize patterns of cortical-subcortical interactions during both reactive and proactive inhibitory control. In Experiment 1, the authors largely replicate previous findings of a cortical negativity related to inhibitory control (both proactive and reactive). In Experiment 2, the authors use direct recordings from the subthalamic nucleus and M1 to show increased coherence between STN and M1 is observed during incongruent trials in the proactive control task (and that the asymmetry of mutual information is directed from STN to M1). In Experiment 3 and Experiment 4, the authors test the specificity assumption of their theory by seeing that motor excitability of hand muscles is reduced when responses are made using another effector (i.e., the feet).

All three reviewers were very positive on the work described and felt that the experiments complement and extend nicely the larger body of work on the basal ganglia and action selection. Whereas most studies in the area are based either on animal controls or (only) patients of Parkinson's disease that undergo surgery for deep brain stimulation. The current study, on the other hand, combines 3 different recording techniques, the two of which are in healthy human subjects and the one in PD patients. Hence, it has the potential to provide a more complete and clearer image on the function of this fundamental brain structure during behavior.

Essential revisions:

1) Selectivity: We had a concern with the nature of the selectivity of the described effects. The authors are arguing that reactive (e.g., stop-signal task) and proactive (e.g., Simon task) rely on a common, non-selective inhibitory mechanism. While Experiment 3 and Experiment 4 show that hand muscles are inhibited during the Simon task when responses are made with the feet, it's not exactly a perfect case for selective inhibition because feet responses are very non-traditional in a computer setting (i.e., the testing environment) and it could very much be that participants are covertly planning responses of with their hands too (the natural/habitual mapping). This doesn't necessarily negate the results of Experiment 3 and Experiment 4, but it does provide a plausible counter hypothesis. A stronger test of the selectivity hypothesis would be to look at laterality of the effects. Experiment 1 has a way of addressing this. Since responses in the Simon task are made with either hand, it should be possible to extract the execution-related responses in each hemisphere. If the inhibition is global and non-selective as the authors propose, then the amplitude of the motor components should be smaller (or changed in an inhibition-consistent way) in the responding hemisphere during incongruent trials (relative to congruent trials). The changes in the lateralized motor components should mirror that observed during the stop-signal experiment during late successful stops. This seems to be one way to validate the selectivity assumption.

2. Analysis of ERD. There were concerns with the limited analysis of Experiment 1. On its own, the experiment is interesting but would not suffice the standards of the present Journal in terms of relevance and novelty. Similar studies have been published before, the latest using prefrontal ECOG and STN LFP (of note, published only 2 days before submission of this manuscript; Zavala, 2018). The ERP of the Simon task should be shown to see the resemblance of the component. This is an unresolved problem with the whole concept of theta in conflict vs. stopping because, the visible ERP can hardly be called an oscillation and does therefore unlikely induce communication through coherence or phase locking. The higher frequency theta shown in the Simon task looks similar to previous moving dot task results (Herz et al.,) but does not include the frequency 2-4 Hz that are typically seen in the strong ERP. Importantly, the negative correlation with the beta ERD should be characterized in more detail. You could compare the conflict/stopping component with the dominant beta ERD component related to movement and calculate correlations between the two phenomena. Theta may be solely suppressed by the earlier occurring beta ERD related to movement onset and not directly associated with inhibitory processes.

3) Links to previous literature: All three reviewers felt that the authors need to make a stronger link to the previous literature. Two reviewers were concerned about the strength of the case that the inhibition-related effects that they see here reflects hyper-direct pathway dynamics. The hyper-direct pathway gets its strongest connections from M1 and prefrontal areas, which according to the present results has no directional influence on STN during conflict. Nothing in the data presented here can reliably distinguish hyper-direct from indirect pathway effects. The long indirect pathway runs through the STN as well and is thought to be the main mechanism by which hyper-direct control signals terminate an unwanted response (see Mallet et al., 2016). In fact, they are really largely distinguished by the origin of their afferent control signals (i.e., cortical vs. striatal/pallidal). The authors even seem to doubt the existence of the indirect pathway altogether in the Discussion section ("While these pathways are still largely theoretical (though anatomical studies support the presence of a hyper-direct pathway […]"). This ignores a very large body of work supporting the existence (both anatomically and physiologically) of these pathways.

The reviewers thought that authors needed to tie their results to computational models in algorithmic or neuronal level, as many of them propose the same (or fairly similar) framework for action selection with transient dynamics, as one of the main conclusions of the current manuscript. See for example the work of Redgrave, Prescott and Gurney, (initially published in 1999), the work of Humphries or Kotaleski among others. This would go a long way to fleshing out the limitations of the study a bit more and be more nuanced in their interpretation of the results.

Finally, one reviewer commented that the Discussion section is weak and by far too self-referential. Findings from animal studies and other disciplines are lacking. How do the authors believe is the red nucleus and brainstem involved in the stopping process, as these are the projection targets of the hyperdirect pathway (see Coudè, 2018, but also the cited Nambu reference). Why was no hyperdirect pathway activity found in the Experiment 2? Are a mix of theta and beta oscillations suggestive of monosynaptic input and disynaptic output in terms of physiology? What is the effect of dopamine on that projection, as beta connectivity is known to be modulated by dopamine (see studies by Litvak and colleagues on LFP-MEG)? Do PD patients have difficulties stopping, due to a loss of hyperdirect axons (Chu, 2017)?

4) Localization: The reviewers had concerns with localization of the conflict/inhibition-related signal. You should provide a better idea of the localization of the ICA component being described in Experiment 1. Having the signal originating from execution areas (e.g., M1) vs. planning areas (e.g., premotor/prefrontal) could dramatically change the interpretation of the results. Figure 2B seems to suggest that the component is more motoric in nature (i.e., onset happens later during failed stops, consistent with the Schall models of inhibitory control). However, if the component arises from pre-SMA or IFG, then it would be more consistent with the hyperdirect pathway (see Comment #3 above).

There were also concerns with localizing the source of the ERD. The beta ERD is not sufficient to localize M1. Beta ERD can be observed from the frontal pole to parietal cortex exceeding S1 to posterior regions. See beta penumbra in Kondylis et al., 2016. Strip locations should be reported. If this cannot be done, the authors should state their interpretations more cautiously and should not include the word M1 or motor cortex. Maybe Frontal or Sensorimotor Cortex may be more suitable. Depending on burr-hole placement, the motor cortex can be up to 5-8 cm posterior, which would extend the electrode paddle lengths. The electrodes could potentially reside more frontal than the authors expect. Please report the used contact pair of the strip for all patients in a table to get at least indirect measures of relative location. Reporting High Frequency Activity (60 – 300 Hz) would also be more convincing for localization.

5) Motivation and background: The manuscript feels like it includes 3 different manuscripts, where the main link between them is understanding the role of the hyper-direct pathway in action selection. It would be nice if the introduction provided a better explanation on why these particular sets of experiments were combined, specifying the benefit and logic of this combined approach.

6. Visualization and analysis: The Time-Frequency plots do not allow a full interpretation of the electrophysiological process. If this was done for statistical purposes, this should include the whole trial period, testing and reporting that no significant effect was found in the baseline period and state and justify the time period of interest for FDR corrected TF stats. This comment is absolutely crucial because dynamic changes in oscillations (esp. slow ones) need to be interpreted in relation to ongoing activity throughout the whole trial. Thus, one would expect standardized time and frequency windows for all TF plots (e.g. 1-30 Hz, -1500 – 2000 ms), including the full frequency width for the Granger analysis in Experiment 2. It is also preferable to use of non-parametric methods if data distributions are not assessed or assessment is not feasible. Electrophysiology across trials can be heavily skewed.

With regard to Experiment 2, the PLV results do not look convincing. The narrow band visualization is statistical double dipping and cherry picking. The lower significant blob at 15 Hz seems to be neglected. It is recommended that the authors to report standard frequency ranges (e.g. 13-30, or 13-20/20-35 Hz). Otherwise, the authors could investigate significant coupling on single trial level and adjust frequency ranges to individual differences (e.g. see Litvak et al., 2011). Reporting absolute PLV (not change to baseline) but relative change in Granger causality (% change to baseline) is confusing and seems post-hoc selective. This should be standardized.

7) Interpreting the results from the patients: Experiment 2 does not make clear that the participants are PD patients. Here, results on beta oscillations are reported that are known to increase dramatically after withdrawal of dopaminergic medication. DBS operations are conducted in the worst clinical state to allow accurate symptom testing. This is not mentioned nor discussed in the manuscript.

In addition, PD patients have motor deficits, such as tremor and bradykinesia that need to be quantified and reported, to allow interpretation of the results. A patient table including Unified Parkinson's Disease Rating Scale (UPDRS-III) scores needs to be included, alongside the symptom dominant hemibody and Parkinsonian subtype (bradykinetic/rigid, equivalent, tremor dominant).

Finally reporting beta coupling in PD patients without mentioning pathophysiology or dopamine is conceptually problematic. A lot of work has been done on corticosubthalamic beta coupling in PD patients.

[Editors' note: further revisions were requested prior to acceptance, as described below.]

Thank you for resubmitting your work entitled "Non-selective inhibition of inappropriate motor-tendencies during response-conflict by a fronto-subthalamic mechanism" for further consideration at *eLife*. Your revised article has been favorably evaluated by Richard Ivry (Senior Editor), Tim Verstynen (Reviewing Editor), and three reviewers.

The manuscript has been improved but we do ask that you address some remaining issues.

One of the reviewers raised one major point that all of us agreed during the discussion was important. The reviewer wrote:

"The authors made an effort to incorporate the suggested changes and I feel that the manuscript has improved significantly from the first revision. From my perspective it is almost ready for publication but not quite there yet. One major point is left that was not addressed sufficiently in the present revision: In Experiment 2, the Simon task induces a reaction time effect through conflictual information on the hand that should be moving. In the present study, as well as in previous studies, the subjects successfully chose the right hands, but did so by taking more time in the incongruent condition, obviously requiring inhibition of premature false responses. However, the authors did not report any neurophysiological evidence of this effect along the fronto-BG axis, e.g. stronger beta coupling in contralateral hemisphere during incongruent vs. congruent trials OR any kind of activity that could underlie the increase in reaction time in the correctly chosen hand. This is surprising and should be discussed.”

In addition to providing a deeper explanation of why no frontal-basal ganglia component is associated with the incongruency behavioral effect in the Simon task, we ask that you consider a formal analysis to evaluate the strength of this null effect (e.g., a Bayes Factor analysis). Given that the lack of association conflicts with (no pun intended) previous findings, we think this would be a nice addition to the analysis.

---

## [Author Response]

Essential revisions:1) Selectivity: We had a concern with the nature of the selectivity of the described effects. The authors are arguing that reactive (e.g., stop-signal task) and proactive (e.g., Simon task) rely on a common, non-selective inhibitory mechanism. While Experiments 3 and 4 show that hand muscles are inhibited during the Simon task when responses are made with the feet, it's not exactly a perfect case for selective inhibition because feet responses are very non-traditional in a computer setting (i.e., the testing environment) and it could very much be that participants are covertly planning responses of with their hands too (the natural/habitual mapping). This doesn't necessarily negate the results of Experiment 3 and Experiment 4, but it does provide a plausible counter hypothesis. A stronger test of the selectivity hypothesis would be to look at laterality of the effects. Experiment 1 has a way of addressing this. Since responses in the Simon task are made with either hand, it should be possible to extract the execution-related responses in each hemisphere. If the inhibition is global and non-selective as the authors propose, then the amplitude of the motor components should be smaller (or changed in an inhibition-consistent way) in the responding hemisphere during incongruent trials (relative to congruent trials). The changes in the lateralized motor components should mirror that observed during the stop-signal experiment during late successful stops. This seems to be one way to validate the selectivity assumption.

To test this alternative hypothesis, we capitalized on the large trial number in our TMS experiments (to which this question pertains) and investigated the proposed laterality effect directly therein. Indeed, if hand-MEP suppression on incongruent trials can be explained by the selective suppression of a purported covertly co-activated hand-response, one would expect a significant laterality effect in the CSE condition differences (since CSE was only measured at the right hand). Therefore, we conducted a control analysis in which we split the trials for each subject by response-side (with each of the resulting four conditions still averaging more than 100 trials per subject) and investigated whether the main effect of CONGRUENCY was modulated by an interaction with RESPONSE SIDE. The newly designed 2x2 rmANOVA showed a significant main effect of congruency (as expected from the t-test result, F(1,29) = 20.5, p =.00009, p-η^2^=.41). Importantly, this was not affected by a significant interaction (F(1,29) = 1.13, p =.3, p-η^2^ =.038). The post-hoc power for this interaction (according to G*Power 3) was >.88. Therefore, we can reject the alternative hypothesis of a significant laterality effect. We have added this analysis into the subsection “Experiment 3.2.: Response-conflict leads to non-selective CSE-suppression”.

2. Analysis of ERD. There were concerns with the limited analysis of Experiment 1. On its own, the experiment is interesting but would not suffice the standards of the present Journal in terms of relevance and novelty. Similar studies have been published before, the latest using prefrontal ECOG and STN LFP (of note, published only 2 days before submission of this manuscript; Zavala, 2018). The ERP of the Simon task should be shown to see the resemblance of the component. This is an unresolved problem with the whole concept of theta in conflict vs. stopping because, the visible ERP can hardly be called an oscillation and does therefore unlikely induce communication through coherence or phase locking. The higher frequency theta shown in the Simon task looks similar to previous moving dot task results (Herz et al.,) but does not include the frequency 2-4 Hz that are typically seen in the strong ERP. Importantly, the negative correlation with the beta ERD should be characterized in more detail. You could compare the conflict/stopping component with the dominant beta ERD component related to movement and calculate correlations between the two phenomena. Theta may be solely suppressed by the earlier occurring beta ERD related to movement onset and not directly associated with inhibitory processes.

We appreciate that Experiment 1 on its own is not sufficiently novel for *eLife*. However, it served as the motivation for the follow-up Experiment 2, Experiment 3, and Experiment 4 (this point is now made clearer in the manuscript), and we believe that it provides complementary results in support of the overall finding.

Regarding the new Zavala et al., study, we were unaware of this publication at the time of our initial submission. We have cited this study in the revised version of the manuscript (Discussion section).

Regarding the Simon task ERP, we have added the ERP to Figure 2. Please note that throughout both the original manuscript as well as the revised version, we never refer to the theta activity as an “oscillation”, exactly because of the issue mentioned here.

Finally, regarding the relationship between beta ERD and the fronto-central activity, we regressed the onset of the beta ERD on individual trials onto the theta activity. We did not find a significant association between the two. However, the single-trial signal-to-noise ratio for the scalp-recorded motor signals is not very impressive. Moreover, this essentially amounts to a null hypothesis test with unknown power. Therefore, while this ‘null-finding’ speaks against the proposed alternative hypothesis proposed here, we would prefer to refrain from adding this analysis to the manuscript. However, if the reviewer or the editors feel strongly about this (or have a better proposal of how exactly to approach this hypothesis), we can of course add it.

3) Links to previous literature: All three reviewers felt that the authors need to make a stronger link to the previous literature. Two reviewers were concerned about the strength of the case that the inhibition-related effects that they see here reflects hyper-direct pathway dynamics. The hyper-direct pathway gets its strongest connections from M1 and prefrontal areas, which according to the present results has no directional influence on STN during conflict. Nothing in the data presented here can reliably distinguish hyper-direct from indirect pathway effects. The long indirect pathway runs through the STN as well and is thought to be the main mechanism by which hyper-direct control signals terminate an unwanted response (see Mallet et al., 2016). In fact, they are really largely distinguished by the origin of their afferent control signals (i.e., cortical vs. striatal/pallidal). The authors even seem to doubt the existence of the indirect pathway altogether in the Discussion section ("While these pathways are still largely theoretical (though anatomical studies support the presence of a hyper-direct pathway […]"). This ignores a very large body of work supporting the existence (both anatomically and physiologically) of these pathways.The reviewers thought that authors needed to tie their results to computational models in algorithmic or neuronal level, as many of them propose the same (or fairly similar) framework for action selection with transient dynamics, as one of the main conclusions of the current manuscript. See for example the work of Redgrave, Prescott and Gurney, (initially published in 1999), the work of Humphries or Kotaleski among others. This would go a long way to fleshing out the limitations of the study a bit more and be more nuanced in their interpretation of the results.Finally, one reviewer commented that the Discussion section is weak and by far too self-referential. Findings from animal studies and other disciplines are lacking. How do the authors believe is the red nucleus and brainstem involved in the stopping process, as these are the projection targets of the hyperdirect pathway (see Coudè, 2018, but also the cited Inase et al., 1999). Why was no hyperdirect pathway activity found in the Experiment 2? Are a mix of theta and beta oscillations suggestive of monosynaptic input and disynaptic output in terms of physiology? What is the effect of dopamine on that projection, as beta connectivity is known to be modulated by dopamine (see studies by Litvak and colleagues on LFP-MEG)? Do PD patients have difficulties stopping, due to a loss of hyperdirect axons (Chu, 2017)?

First, we agree with the reviewers that the relative roles of the hyper-direct and indirect pathways in our data pattern is largely speculative, given the nature of our experiments. We previously took some liberties in the discussion to this effect, based on the fact that global suppression of motor cortex (as found in our Experiment 3 and Experiment 4) have been largely attributed to the hyperdirect pathway (cf. Majid et al.,2013; Jahanshahi et al., 2015). However, in the revised discussion, we have moved away from strong statements regarding the relative roles of the hyper-direct vs. the indirect pathway, and have stuck closer to our current data.

Second, we in no way meant to imply that we do not believe there to be evidence for an indirect pathway. This was a misunderstanding of some ambiguous phrasing. This statement was meant to convey that the hyper-direct pathway in humans is still largely theoretical. We have changed the phrasing of that statement top hopefully avoid confusion (Discussion section).

Third, we have added a section regarding the computational work and how it pertains to our current study (Discussion section).

Lastly, however, in the spirit of sticking closer to what our current data can (and cannot) show, we chose not to add a discussion of the role of the red nucleus and the brain stem. While interesting, we did not record from these regions in our current study, and a discussion of these structures would be even more speculative than our previous assertions about hyper-direct vs. indirect pathway. We hope that this makes sense.

4) Localization: The reviewers had concerns with localization of the conflict/inhibition-related signal. You should provide a better idea of the localization of the ICA component being described in Experiment 1. Having the signal originating from execution areas (e.g., M1) vs. planning areas (e.g., premotor/prefrontal) could dramatically change the interpretation of the results. Figure 2B seems to suggest that the component is more motoric in nature (i.e., onset happens later during failed stops, consistent with the Schall models of inhibitory control). However, if the component arises from pre-SMA or IFG, then it would be more consistent with the hyperdirect pathway (see Comment #3 above).There were also concerns with localizing the source of the ERD. The beta ERD is not sufficient to localize M1. Beta ERD can be observed from the frontal pole to parietal cortex exceeding S1 to posterior regions. See beta penumbra in Kondylis et al., 2016. Strip locations should be reported. If this cannot be done, the authors should state their interpretations more cautiously and should not include the word M1 or motor cortex. Maybe Frontal or Sensorimotor Cortex may be more suitable. Depending on burr-hole placement, the motor cortex can be up to 5-8 cm posterior, which would extend the electrode paddle lengths. The electrodes could potentially reside more frontal than the authors expect. Please report the used contact pair of the strip for all patients in a table to get at least indirect measures of relative location. Reporting High Frequency Activity (60 – 300 Hz) would also be more convincing for localization.

To address the first point, we have performed a dipole-localization analysis of the ICs underlying the analyses in Experiment 1. While we are generally skeptical of the potential of inverse source solutions of scalp EEG data to test precise anatomical hypotheses, these data may indeed be helpful in understanding whether there is any laterality to the sources underlying Experiment 1 (which would indicate a motor source) or whether they are more likely originate from the fronto-central midline structures (more in line with a control-related function).

In short, the inverse dipole solutions clearly support a midline source. Whereas there is some variability (yellow bubble) in the ventral-dorsal dimension across individual subjects’ components, there is almost no variability in the lateral dimension (Author response image 1, right plot). This clearly suggests a midline source. That is also in line with previous simultaneous EEG-fMRI work showing that fronto-central activity following stop-signals correlates with the BOLD response in medial wall structures (pre-SMA and ACC, cf. Enriquez-Geppert et al., 2010).

Lastly, if the fronto-central EEG signal were indeed ‘motoric’ in nature (i.e., one reflective of motor execution rather than inhibitory control), it would be hard to explain why there is no amplitude difference between successful stop-trials (where no response is made) and failed stop-trials (where a response is made). Instead, the differentiating factor between those two conditions is clearly the earlier onset on successful trials, which is the exact prediction that would be made for an inhibitory process in the SST.

**Author response image 1. respfig1:** Dipole analysis of the ICs selected to represent the fronto-central P3 in the stop-signal task, averaged across subjects. Yellow bubble = standard deviation.

To address the question regarding the specificity and localization of the “M1” beta-suppression signals in Experiment 2, we have now quantified activity not just at the selected “M1” contacts, but also at the adjacent ones. Since our electrode strip had four contacts (resulting in three possible bipolar montages), we plotted the immediately anterior montage for all subjects in which either the most posterior or the middle montage was selected, and we also plotted the immediately posterior montage for all subjects in which the most anterior or middle montage was selected. Furthermore, we also plotted the selected montage for each subset of subjects to match the subjects that had a posterior or anterior montage in addition to the selected montage. This was done to avoid biasing our selected montage towards showing a cleaner signal because of a larger sample size for the selected montage. In summary, these analyses clearly show that the beta-band suppression is localized to the selected montage. We have added these plots to Figure 4 in the revised version of the manuscript (also cf. additional Materials and methods section).

5) Motivation and background: The manuscript feels like it includes 3 different manuscripts, where the main link between them is understanding the role of the hyper-direct pathway in action selection. It would be nice if the introduction provided a better explanation on why these particular sets of experiments were combined, specifying the benefit and logic of this combined approach.We have edited the Introduction to reflect the link between the individual experiments that underlie this study. It now contains what is essentially a description of the actual “history” of this investigation: Experiment 1 was used to generate first evidence towards the fact that response-conflict and action-stopping may involve overlapping neural activity, which motivated Experiment 2. The findings of Experiment 2 then motivated Experiments 3.1. and 3.2.6. Visualization and analysis: The Time-Frequency plots do not allow a full interpretation of the electrophysiological process. If this was done for statistical purposes, this should include the whole trial period, testing and reporting that no significant effect was found in the baseline period and state and justify the time period of interest for FDR corrected TF stats. This comment is absolutely crucial because dynamic changes in oscillations (esp. slow ones) need to be interpreted in relation to ongoing activity throughout the whole trial. Thus, one would expect standardized time and frequency windows for all TF plots (e.g. 1-30 Hz, -1500 – 2000 ms), including the full frequency width for the Granger analysis in Experiment 2. It is also preferable to use of non-parametric methods if data distributions are not assessed or assessment is not feasible. Electrophysiology across trials can be heavily skewed.With regard to Experiment 2, the PLV results do not look convincing. The narrow band visualization is statistical double dipping and cherry picking. The lower significant blob at 15 Hz seems to be neglected. It is recommended that the authors to report standard frequency ranges (e.g. 13-30, or 13-20/20-35 Hz). Otherwise, the authors could investigate significant coupling on single trial level and adjust frequency ranges to individual differences (e.g. see Litvak et al., 2011). Reporting absolute PLV (not change to baseline) but relative change in Granger causality (% change to baseline) is confusing and seems post-hoc selective. This should be standardized.

In line with the proposal, we have extended the depicted time periods for all time-frequency analyses that include low-frequency activity. However, a 3.5s window of 1500 to 2000ms, as proposed here, is excessively long. While individual groups may favor such windows for slow-paced tasks (e.g., dot-probe tasks), this is highly unusual for tasks with faster trial timing (where time periods of that length will include activity from preceding or subsequent trials). In their landmark paper on the role of frontal theta in cognitive control, Cavanagh and Frank (TiCS, 2014) present windows of 1 second around events for tasks involving conflict, errors, surprise, or novelty (starting at 250ms pre event and last for 750ms after the event). We have matched this window length in our revised version by plotting the data from - 300 to 700ms with respect to the event in all analyses that focused on low frequencies (Figure 2C and D).

Regarding the PLV line-graphs: The narrow-band analysis is a visualization of the relevant frequency bands for the two conditions separately (which cannot be inferred from the full-spectrum difference plot). Presenting these data in a full-spectrum plot is not very informative. We do, however, agree that a separate statistical test on these line-graphs may be construed as unnecessary. Therefore, we have removed those tests from the line graphs to make clear that these graphs are merely presented to visualize the individual conditions at the frequencies that were found significant in the full-spectrum analysis.

Regarding the baseline correction, the PLV estimates were indeed baseline corrected. However, this was not explicated in the previous Materials and methods section, which was now amended to that effect.

Finally, regarding the Granger causality analysis, there are two reasons why that analysis was performed on the beta-range only. First, we had an a priori hypothesis generated from the PLV analyses, where only beta-band activity showed PLV condition differences in the full-spectrum analysis. Including additional frequencies in the Granger causality analysis would inappropriately inflate type-I error probability. The more important reason, however, is that Granger-causality analyses involve a trade-off between optimal frequency resolution, model order, and model stability over time. This trade-off varies between frequencies (Cohen, 2014). Hence, a full-spectrum analysis of Granger coefficients with the same model order is inappropriate, making it impossible to sensibly compare coefficients across frequencies. We have added an explanation of this to the revised version of the manuscript.

7) Interpreting the results from the patients: Experiment 2 does not make clear that the participants are PD patients. Here, results on beta oscillations are reported that are known to increase dramatically after withdrawal of dopaminergic medication. DBS operations are conducted in the worst clinical state to allow accurate symptom testing. This is not mentioned nor discussed in the manuscript.In addition, PD patients have motor deficits, such as tremor and bradykinesia that need to be quantified and reported, to allow interpretation of the results. A patient table including Unified Parkinson's Disease Rating Scale (UPDRS-III) scores needs to be included, alongside the symptom dominant hemibody and Parkinsonian subtype (bradykinetic/rigid, equivalent, tremor dominant).Finally reporting beta coupling in PD patients without mentioning pathophysiology or dopamine is conceptually problematic. A lot of work has been done on corticosubthalamic beta coupling in PD patients.

The fact that Experiment 2 involves patients and intraoperative recordings is mentioned as early as the Introduction. It is then reiterated at the very beginning of the results section for Experiment 2, as well as in the Materials and methods section for Experiment 2 of the original manuscript. In the revised manuscript, we have now also added mention of this into the Discussion.

With regards to detailed symptom information, we have added a detailed table to that effect into the revised version of the manuscript.

[Editors' note: further revisions were requested prior to acceptance, as described below.]

The manuscript has been improved but we do ask that you address some remaining issues.One of the reviewers raised one major point that all of us agreed during the discussion was important. The reviewer wrote:"The authors made an effort to incorporate the suggested changes and I feel that the manuscript has improved significantly from the first revision. From my perspective it is almost ready for publication but not quite there yet. One major point is left that was not addressed sufficiently in the present revision: In Experiment 2, the Simon task induces a reaction time effect through conflictual information on the hand that should be moving. In the present study, as well as in previous studies, the subjects successfully chose the right hands, but did so by taking more time in the incronguent condition, obviously requiring inhibition of premature false responses. However, the authors did not report any neurophysiological evidence of this effect along the fronto-BG axis, e.g. stronger β coupling in contralateral hemisphere during incongruent vs. congruent trials OR any kind of activity that could underlie the increase in reaction time in the correctly chosen hand. This is surprising and should be discussed.”In addition to providing a deeper explanation of why no frontal-basal ganglia component is associated with the incongruency behavioral effect in the Simon task, we ask that you consider a formal analysis to evaluate the strength of this null effect (e.g., a Bayes Factor analysis). Given that the lack of association conflicts with (no pun intended) previous findings, we think this would be a nice addition to the analysis.

As a general remark prior to our specific response to this issue, we want to point out that the assessment that “no frontal-basal ganglia component is associated with the incongruency behavioral effect in the Simon task” is not entirely accurate. The frontal EEG signature in Experiment 1 clearly shows such an association (cf. correlation analysis in Figure 2C, right hand plot).

However, we agree that it is somewhat surprising that the STN-based analysis did not show the same association for the responding hand that it shows for the non-responding hand. In the revised version of the manuscript, we have addressed this as follows:

1) In line with the reviewer’s suggestion, we have added Bayes factor analyses to Figure 5—figure supplement 1 (the figure that shows the null effect of conflict on STN-M1 connectivity). This analysis shows that there is positive evidence (BF > 3) for the null hypothesis during most parts of the spectrum; though at some part of the spectrum, the effects are inconclusive (BF between 0 and 3). Importantly, during the period during which the ipsilateral analysis yielded significant differences (~200-300ms), there is consistent positive evidence for the null hypothesis (BF > 3).

2) To contextualize this finding and discuss the considerations proposed by the reviewer, we have added the following paragraph to the Discussion section of the revised manuscript:

“One notable aspect about our study is the absence of connectivity between STN and contralateral M1 during response-conflict in Experiment 2. Experiments 1 and 3 clearly show that there is an association between inhibitory signatures at scalp- and motor-system level and the degree of conflict-related behavioral slowing observed at the correct effector. […] Another alternative explanation is that the simultaneous excitation and inhibition of contralateral M1 during response-conflict could have mutually countermanding effects on M1, which could obscure the degree to which inhibitory influence of STN on M1 is measurable using the connectivity techniques used here. Future studies that simultaneously image several basal ganglia nuclei may be able to resolve this question.”